Modelling the habitat preference of two key *Sphagnum* species in a poor fen as controlled by
capitulum water content
Jinnan Gong[1], Nigel Roulet[2], Steve Frolking[1,3], Heli Peltola[1], Anna M. Laine[1,4], Nicola
Kokkonen[1], Eeva-Stiina Tuittila[1]
[1] School of Forest Sciences, University of Eastern Finland, P.O. Box 111, FI-80101 Joensuu,
Finland
[2] Department of Geography, McGill University and Centre for Climate and Global Change
Research, Burnside Hall, 805 rue Sherbrooke O., Montréal, Québec H3A 2K6
[3] Institute for the Study of Earth, Oceans, and Space, and Department of Earth Sciences,
University of New Hampshire, Durham, NH 03824, USA
[4] Department of Ecology and Genetics, University of Oulu, P.O. Box 3000, FI-90014, Oulu,
Finland
**Abstract**
Current peatland models generally treat vegetation as static, although plant community structure
is known to alter as a response to environmental change.  Because the vegetation structure and
ecosystem functioning are tightly linked, realistic projections of peatland response to climate
change require including vegetation dynamics in ecosystem models. In peatlands, *Sphagnum*
mosses are key engineers. Moss community composition primarily follows habitat moisture
conditions. The species known preference along the prevailing moisture gradient might not
directly serve as a reliable predictor for future species compositions as water table fluctuation is
likely to increase. Hence, modelling the mechanisms that control the habitat preference of
*Sphagna* is a good first step for modelling community dynamics in peatlands. In this study, we
developed the Peatland Moss Simulator (PMS), simulating community dynamics of the peatland
moss layer. PMS is a process-based model that employs a stochastic, individual-based approach
simulating competition within peatland moss layer based on species differences in functional
traits. At the shoot-level, growth and competition were driven by net photosynthesis, which was
regulated by hydrological processes via capitulum water content. The model was tested by
predicting the habitat preferences of *S. magellanicum* and *S. fallax*, two key species representing
dry (hummock) and wet (lawn) habitats in a poor fen peatland (Lakkasuo, Finland). PMS
successfully captured the habitat preferences of the two *Sphagnum* species, based on observed
variations in trait properties. Our model simulation further showed that the validity of PMS
depended on the interspecific differences in capitulum water content being correctly specified.
Neglecting the water content differences led to the failure of PMS to predict the habitat
preferences of the species in stochastic simulations. Our work highlights the importance of

capitulum water content to the dynamics and carbon functioning of *Sphagnum* communities in peatland ecosystems. Studies of peatland responses to changing environmental conditions thus need to include capitulum water processes as a control on moss community dynamics. Our PMS model could be used as an elemental design for the future development of dynamic vegetation models for peatland ecosystems.

**Keywords**: *Sphagnum* moss; capitulum water content; competition; peatland community dynamics; process-based modelling; moss traits; Peatland Moss Simulator (PMS)

## 1.Introduction

Peatlands have important roles in the global carbon cycle as they store about 30% of the world's soil carbon (Gorham, 1991; Hugelius et al., 2013). Environmental changes, like climate warming and land-use changes, are expected to impact the carbon functioning of peatland ecosystems (Tahvanainen, 2011). Predicting the functioning of peatlands under environmental changes requires models to quantify the interactions among ecohydrological, ecophysiological and biogeochemical processes. These processes are known to be strongly regulated by vegetation (Riutta et al. 2007; Wu and Roulet, 2014), which can change over decadal time scales under changing hydrological conditions (Tahvanainen, 2011). Peatland models have generally considered vegetation structure unrealistically as static component (e.g. Frolking et al., 2002; Wania et al., 2009). The recent regional-scale peatland model developed by Chaudhary et al. (2017) includes dynamic vegetation shifts among a single moss plant functional type (PFT) and four vascular PFTs but to support realistic predictions on peatland functioning and global biogeochemical cycles the mechanisms that drive changes in moss community structure need to be identified and integrated with ecosystem processes.

A major fraction of peatland biomass is formed by *Sphagnum* mosses (Hayward and Clymo, 1983; Vitt, 2000). Although individual *Sphagnum* species often have narrow habitat niches (Johnson et al., 2015), different *Sphagnum* species replace each other along water table gradient and therefore, as a genus, spread across a wide range of water table conditions (Rydin and McDonald, 1985; Andrus et al. 1986; Rydin, 1993; Laine et al. 2009). The species composition of the *Sphagnum* community strongly affects ecosystem processes such as carbon sequestration and peat formation through interspecific variability in species traits such as photosynthetic potential and litter quality (Clymo, 1970; O'Neill, 2000; Vitt, 2000; Turetsky, 2003). The *Sphagnum* biomass and litter production gradually raises the moss carpet, which feeds back into the species composition (Robroek et al. 2009). Hence, modelling the moss community dynamics is fundamental for predicting temporal changes of peatland vegetation. As the distribution of *Sphagnum* species primarily follows the variability in peatland water table

(Andrus 1986; Väliranta et al. 2007), modelling the habitat preference of *Sphagnum* species along a moisture gradient could be a good first step for predicting moss community dynamics (Blois et al., 2013).

For a given *Sphagnum* species, the optimal habitat represents the environmental conditions for it to achieve higher rates of net photosynthesis and shoot elongation than its peers (Titus & Wagner, 1984; Rydin & McDonald, 1985; Rydin, 1997; Robroek et al., 2007a; Keuper et al., 2011). Capitulum water content and water storage, which is determined by the balance between the evaporative loss and water gains from capillary rise and precipitation, represents one of the most important controls on net photosynthesis (Titus & Wagner, 1984; Murray et al. 1989; Van Gaalen et al. 2007; Robroek et al., 2009). To quantify the water processes in mosses, hydrological models have been developed to simulate the water movement between moss carpet and the peat underneath, as regulated by the variations in meteorological conditions and energy balance (Price, 2008; Price and Waddington, 2010). On the other hand, experimental work has addressed the species-specific responses of net photosynthesis to changes in capitulum water content (Titus & Wagner, 1984; Hájek and Beckett, 2008; Schipperges and Rydin, 2009) and light intensity (Rice et al., 2008; Laine et al., 2011; Bengtsson et al., 2016). Net photosynthesis and hydrological processes are linked via capitulum water retention, which controls the response of capitulum water content to water potential changes (Jassey & Signarbieux, 2019). However, these mechanisms have not been integrated with ecosystem processes in modelling.

Along with the capitulum water processes, modelling the habitat preference of *Sphagna* requires quantification of the competition among mosses, i.e., the "race for space" (Rydin, 1993; Rydin, 1997; Robroek et al., 2007a; Keuper et al., 2011): *Sphagnum* shoots can form new capitula and spread laterally, if there is space available. This reduces or eliminates the light source for any plant that is buried by its peers (Robroek et al. 2009). As the competition occurs between neighboring shoots, its modelling requires downscaling water-energy processes from the ecosystem to the shoot level. For that, *Sphagnum* competition needs to be modelled as spatial processes, considering that spatial coexistence and the variations of functional traits among shoot individuals may impact the community dynamics (Bolker et al., 2003; Amarasekare, 2003). However, coexistence generally relies on simple coefficients to describe the interactions among individuals (e.g. Czárán and Iwasa, 1998; Anderson and Neuhauser, 2000; Gassmann et al., 2003; Boulangeat et al., 2018), thus being decoupled from environmental fluctuation or the stochasticity of biophysiological processes.

This study aims to develop and test a model, the Peatland Moss Simulator (PMS), to simulate community dynamics within the peatland moss layer that results in realistic habitat preference of *Sphagnum* species along a moisture gradient. In PMS, community dynamics is driven by *Sphagnum* photosynthesis. Photosynthesis in turn is regulated by capitulum water retention through capitulum moisture content. Therefore, we hypothesize that water retention of the

capitula is the mechanism driving moss community dynamics. We test the model validity using
data from an experiment based on two *Sphagnum* species with different positions along moisture
gradient in the same peatland site. If our hypothesis holds, the model will (1) correctly predict
the competitiveness of the two species in wet and dry habitats; and (2) fail to predict
competitiveness if the capitulum water retention and water content of the two species are not
correctly specified.

## 2. Materials and methods

### 2.1 Study site

The peatland site being modelled is located in Lakkasuo, Orivesi, Finland (61° 47' N; 24° 18'
E). The site is a poor fen fed by mineral inflows from a nearby esker (Laine et al 2004). Most of
the site is formed by lawns dominated by *Sphagnum recurvum* complex (*Sphagnum fallax*,
accompanied by *Sphagnum flexuosum* and *Sphagnum angustifolium*) and *Sphagnum papillosum*.
Less than 10% of surface is occupied by hummocks, with *Sphagnum magellanicum* and
*Sphagnum fuscum,* being 15-25 cm higher than the lawn surfaces. Both microforms are covered
by continuous *Sphagnum* carpet with a sparse cover of vascular plants (projection cover of *Carex*
12% on average), which spread homogeneously over the topography. The annual mean water
table was 15.6 ± 5.0 cm deep at lawn surface (Kokkonen et al., 2019). More information about
the site can be found in Kokkonen et al. (2019).

### 2.2 Model outline

The Peatland Moss Simulator (PMS) is a process-based, stochastic model, which simulates the
temporal dynamics of *Sphagnum* community as driven by variations in precipitation, irradiation,
and energy flow with individual-based interactions (Fig. 1). In PMS, the studied ecosystem is
seen as a dual-column system consisting of hydrologically connected habitats of hummocks and
lawns (community environment in Fig. 1). For each habitat type, the community area is
downscaled to two-dimensional cells representing the scale of individual shoots (i.e. 1 cm$^2$).
Each grid cell can be occupied by one capitulum from a single *Sphagnum* species. The
community dynamics, i.e. the changes in species abundances, are driven by the growth and
competition of *Sphagnum* shoots at the grid-cell level (Module I in Fig. 1). These processes were
regulated by the grid-cell-specific conditions of water and energy (Module II in Fig. 1), which
are derived from the community environment (Module III in Fig. 1).
In this study, we focused on developing Module I and II (Section 2.3) and employed an
available soil-vegetation-atmosphere transport (SVAT) model (Gong et al., 2013a, 2016) to
describe the water-energy processes for Module III (Appendix A). We assumed that the temporal
variation in water table was similar in lawns and hummocks, and the hummock-lawn differences
in water table (*dWT* in Fig. 1) followed their difference in surface elevations (Wilson, 2012). At
the grid cell level, the photosynthesis of capitula drove the biomass growth and elongation of
shoots, which led to the competition between adjacent grid cells. The net photosynthesis rate was
controlled by capitulum water content ($W_{cap}$), which was defined by the capitulum water
retention in relation to water potential (*h*) (Section 2.4). The values for functional traits that
regulate the growth and competition processes were randomly selected within their normal
distribution measured in the field (Section 2.4). Unknown parameters that related the lateral
water flows of the site are estimated using a machine-learning approach (Section 2.5). Finally,
Monte-Carlo simulation was used to support the analysis on the habitat preferences of *Sphagnum*
species and hypothesis tests (Section 2.6). The list of used symbols is given in Table 1.

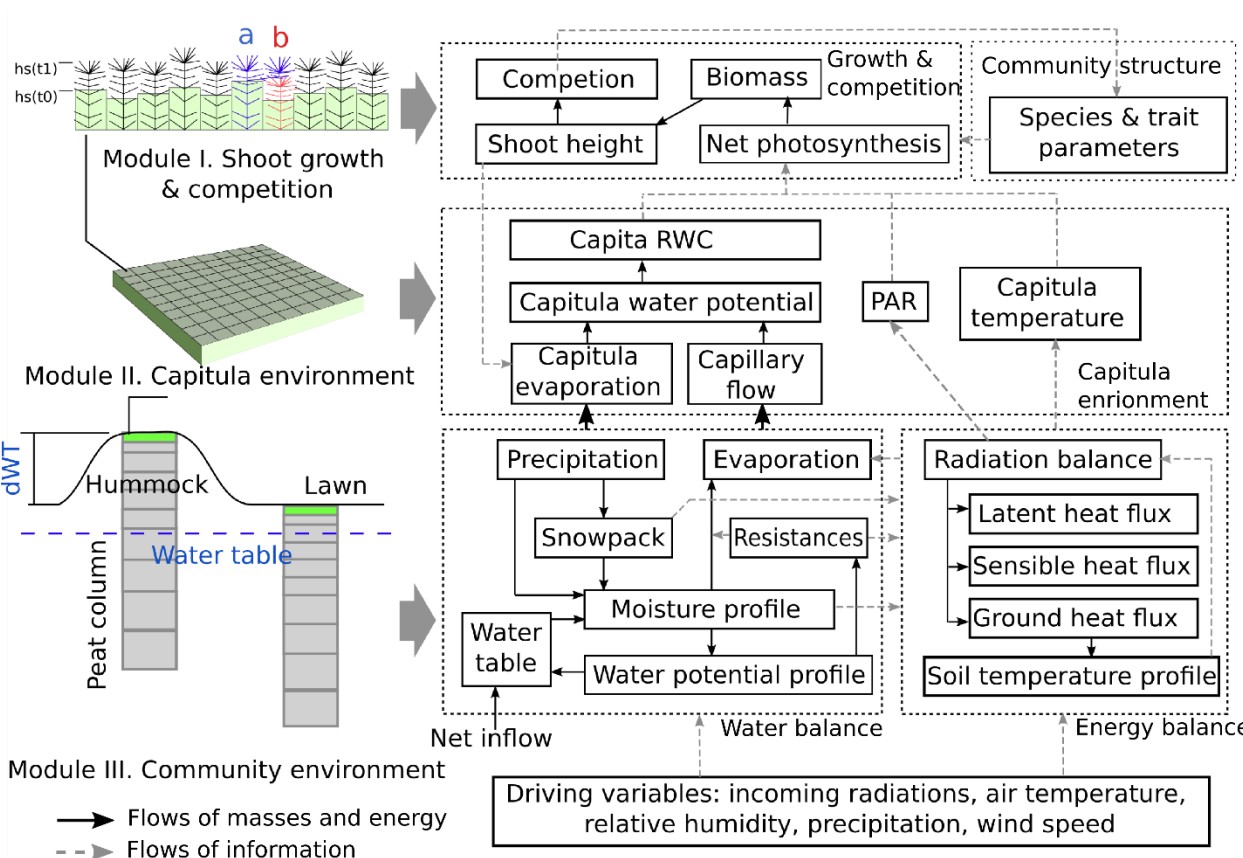


Fig. 1 Framework of Peatland Moss Simulator (PMS).

**2.3 Model development**
**2.3.1 Calculating shoot growth and competition of Sphagnum mosses (Module I)**

*Calculation of Sphagnum growth*

To model grid cell biomass production and height increment, we assumed that capitula were the main parts of shoots responsible for photosynthesis and production of new tissues, instead of the stem sections underneath. We employed a hyperbolic light-saturation function (Larcher, 2003) to calculate the net photosynthesis, which was parameterized based on empirical measurements made from the target species collected from the study site (see Appendix B for materials and methods):

$$A_{20} = \left( \frac{Pm_{20}*PPFD}{\alpha_{PPFD}+PPFD} - Rs_{20} \right) * B_{cap} \tag{1}$$

where subscript 20 denotes the variable value measured at 20 °C; $Rs$ is the mass-based respiration rate ($\mu$mol g$^{-1}$ s$^{-1}$); $Pm$ is the mass-based rate of maximal gross photosynthesis ($\mu$mol g$^{-1}$ s$^{-1}$); $PPFD$ is the photosynthetic photon flux density ($\mu$mol m$^{-2}$ s$^{-1}$); $B_{cap}$ is the capitulum biomass; and $\alpha_{PPFD}$ is the half-saturation point ($\mu$mol m$^{-2}$ s$^{-1}$) for photosynthesis.

By adding multipliers for capitula water content ($f_W$) and temperature ($f_T$) to Eq. (1), the net photosynthesis rate $A$ ($\mu$mol m$^{-2}$ s$^{-1}$) was calculated as following:

$$A = \left[ \frac{Pm_{20}*PPFD}{\alpha_{PPFD}+PPFD} f_T(T) - Rs_{20}f_R(T) \right] * B_{cap} * f_W(W_{cap}) \tag{2}$$

where $f_W(W_{cap})$ describes the responses of $A$ to capitulum water content, $W_{cap}$; $f_T(T)$ describes the responses of $Pm$ to capitulum temperature $T$ (Korrensalo et al., 2017). $f_W(W_{cap})$ was estimated based on the empirical measurements (Appendix B; see Section 2.4). The temperature response $f_R(T)$ is a Q$_{10}$ function that describes the temperature sensitivity of $Rs$ (Frolking et al., 2002):

$$f_R(T) = Q_{10}^{(T-T_{opt})/10} \tag{3}$$

where Q$_{10}$ is the sensitivity coefficient; $T$ is the capitulum temperature (°C); $T_{opt}$ (20 °C) is the reference temperature of respiration.

The response of $A$ to $W_{cap}$ ($f_W(W_{cap})$, Eq. 2) was described as a second-order polynomial function):

$$f_W(W_{cap}) = a_{W0} + a_{W1} * W_{cap} + a_{W2} * W_{cap}^2 \tag{4}$$

where $a_{W0}$, $a_{W1}$ and $a_{W2}$ are coefficients.

Plants can store carbohydrates as nonstructural carbon (NSC, e.g. starch and soluble sugar) to support fast growth in spring or post-stress periods, like after drought events (Smirnoff et al., 1992; Martínez-Vilalta et al., 2016; Hartmann and Trumbore, 2016). We linked the production of shoot biomass to the immobilization of NSC storage (modified from Eq. 10 in Asaeda and Karunaratne, 2000). The change in NSC storage depends on the balance between net photosynthesis and immobilization:

$M_B = s_{imm} * \text{NSC} * k_{imm}\alpha_{imm}^{T-20}$

194     (5)

$\partial\,NSC/\partial\,t = A - M_B, NSC\epsilon[0, NSC_{max}]$                                    (6)
where $M_B$ is the immobilized NSC to biomass production during a time step (g); $k_{imm}$ is the
specific immobilization rate (g g$^{-1}$) (Asaeda and Karunaratne 2000); $\alpha_{imm}$ is the temperature
constant; $s_{imm}$ is the multiplier for temperature threshold, where $s_{imm} = 1$ when $T > 5$ ºC but $s_{imm} =$
0 if $T \le 5$ ºC. $NSC_{max}$ is the maximal NSC concentration in *Sphagnum* biomass (Turetsky et al.,
2008). Timing of growth is controlled by a temperature threshold and NSC availability. Growth
occurs when $T > 5$ ºC and NSC is above zero. The dynamics of NSC storage are related to WC
through net photosynthesis.
The increase in shoot biomass drove the shoot elongation:
$\partial\,Hc/\partial\,t = \dfrac{M_B}{H_{spc}S_c}$                                    (7)
where $Hc$ is the shoot height (cm); $H_{spc}$ is the biomass density of *Sphagnum* stems (g m$^{-2}$ cm$^{-1}$)
and $S_c$ is the area of a cell (m$^2$).

*Calculation of Sphagnum competition and community dynamics*
To simulate the competition among *Sphagnum* shoots, we first compared $Hc$ of each grid cell
(source grid cell, i.e. grid cell *a* in Fig. 1) to its four neighboring cells and marked the one with
lowest position (e.g. grid cell *b* in Fig. 1) as the target of spreading. The spreading of shoots from
a source to a target grid cell occurred when the following criteria were fulfilled: i) the height
difference between source and target grid cells exceeded a threshold value; ii) NSC accumulation
in the source grid cell was large enough to support the growth of new capitula in the target grid
cell; iii) the capitula in the source grid cell can split at most once per year.
The threshold of height difference in rule i) was set equal to the mean diameter of capitula in
the source cell, based on the assumption that the shape of a capitulum was spherical. When
shoots spread, the species type and model parameters in the target grid cell were overwritten by
those in the source grid cell, assuming the mortality of shoots originally in the target cell. During
the spreading, NSC storage was transferred from the source cell to the target cell to form new
capitula. In cases where spreading did not take place, establishment of new shoots from spores
could maintain the continuity of *Sphagnum* carpet at the site. During the establishment from
spores, which was rare and occurred during the first years of simulation, the traits of *Sphagnum*
species were randomized within their normal distribution measured in the field.

## 226  2.3.2 Calculating grid cell-level dynamics of environmental factors (Module II)

Module II computes grid-cell values of $W_{cap}$, *PPFD* and *T* for Module I. The cell-level *PPFD*
and *T* were assumed to be equal to the community means, which were solved by the SVAT
scheme in Module III (Appendix A.). The community level evaporation rate (*E*) was partitioned
to cell-level ($E_i$) as following:
$$E_i = E * \left( \frac{Sv_i}{r_{bulk,i}} \right) \Big/ \sum \left( \frac{Sv_i}{r_{bulk,i}} \right) \tag{8}$$
where $r_{bulk,i}$ is the bulk surface resistance of cell *i*, which is as a function ($r_{bulk,i} = fr(h_i)$) of grid-
cell-based water potential $h_i$, capitulum biomass ($B_{cap}$) and shoot density ($D_S$) based on the
empirical measurements (Appendix B); $Sv_i$ was the evaporative area, which was related to the
height differences among adjacent grid cells:
$$Sv_i = Sc_i + lc \sum_j (Hc_i - Hc_j)$$
$\qquad$ (9)
where *lc* is the width of a grid cell (cm); and subscript *j* denotes the four-nearest neighbouring
grid cells. In this way, changes in the height difference between the neighboring shoots feeds
back to affect the water conditions of the grid cells, via alteration of the evaporative surface area.
$\qquad$ The grid cell-level changes in capitula water potential ($h_i$) were driven by the balance between
the evaporation ($E_i$) and the upward capillary flow to capitula:
$$\partial h_i = \frac{K_m}{C_i} \left[ \frac{(h_i - h_m)}{0.5z_m} - 1 - E_i \right] \tag{10}$$
where $h_m$ is the water potential of the living moss layer, solved in Module III (Appendix A.); $z_m$
is the thickness of the living moss layer ($z_m$=5 cm); $K_m$ is the hydraulic conductivity of the moss
layer and that is set to be the same for each grid cell; $C_i$ is the cell-level specific water uptake
capacity ($C_i = \partial W_{cap,i}/\partial h_i$). $\partial W_{cap,i}/\partial h_i$ could be derived from the capitulum water retention
function $h_i = f_h(W_{cap})$. $W_{cap}$ can be then calculated from the estimated from $h_i$ and affect the
calculation of net photosynthesis through $f_W(W_{cap})$ (Eq. 2).

## 251  2.4 Model parameterization

*Selection of Sphagnum species*
We chose *S. fallax* and *S. magellanicum*, which form 63% of total plant cover at the study site at
Lakkasuo (Kokkonen et al., 2019), as the target species representing the lawn and hummock
habitats respectively. These species share a similar niche along the gradients of soil pH and
nutrient richness (Wojtuń et al., 2003), but are discriminated by their preferences of water table
level (Laine et al., 2004). While *S. fallax* is commonly found close to the water table (Wojtuń et
al., 2003), *S. magellanicum* can occur along a wider range of a dry-wet gradient, from
intermediately wet lawns up to dry hummocks (Rice et al., 2008; Kyrkjeeide, et al., 2016;
Korresalo et al., 2017). The transition from *S. fallax* to *S. magellanicum* along the wet-dry
gradient thus indicates the decreasing competitiveness of *S. fallax* against *S. magellanicum* with
a lowering water table.
*Parameterization of morphological traits, net photosynthesis and capitulum water retention*
We empirically quantified the morphological traits capitulum density ($D_S$, shoots cm$^{-2}$), biomass
of capitula ($B_{cap}$, g m$^{-2}$), biomass density of living stems ($H_{spc}$, g cm$^{-1}$ m$^{-2}$), net photosynthesis
parameters ($Pm_{20}$, $Rs_{20}$ and $\alpha_{PPFD}$) and the water retention properties (i.e., $f_h(W_{cap})$ and $fr(h)$, Eqs.
8 and 10) for the two *Sphagnum* species (see Appendix B for methods). The values (mean ± SD)
of the morphological parameters, the photosynthetic parameters and polynomial coefficients
($a_{W0}$, $a_{W1}$ and $a_{W2}$, Eq. 3) are listed in Table 2. For each parameter, a random value was
initialized for each cell based on the measured means and SD, assuming the variation of
parameter values is normally distributed.
We noticed that the fitted $f_W(W_{cap})$ was meaningful when $W_{cap}$ was below the optimal water
content for photosynthesis ($W_{opt}$ = -0.5 $a_{W1}$/ $a_{W2}$). If $W_{cap}$ > $W_{opt}$, photosynthesis decreased
linearly with increasing $W_{cap}$, as being limited by the diffusion of $CO_2$ (Schipperges and Rydin,
1998). In that case, $f_W(W_{cap})$ was calculated following Frolking et al. (2002):
$$f_W(W_{cap}) = 1 - 0.5 \frac{W_{cap} - W_{opt}}{W_{max} - W_{opt}}$$

277          (11)

where $W_{max}$ is the maximum water content of capitula.
It is known that $W_{max}$ is around 25-30 g g$^{-1}$ (e.g. Schipperges and Rydin, 1998), or about 0.31 -
0.37 cm$^3$ cm$^{-3}$ in term of volumetric water content (assuming 75 g m$^{-2}$ capitula biomass and 0.6
cm height of capitula layer). This range is broadly lower than the saturated water content of moss
carpet (> 0.9 cm$^3$ cm$^{-3}$, McCarter and Price, 2014). Consequently, we used the following
equation to convert volumetric water content to capitula RWC, when $h_i$ was higher than the
boundary value of -10$^4$ cm:
$$W_{cap} = min(W_{max}, \theta_m/(H_{cap} * B_{cap} * 10^{-4}))$$
286          (12)

where $W_{max}$ is the maximum water content that set to 25 g g$^{-1}$ for both species; $\theta_m$ is the
volumetric water content of moss layer; $H_{cap}$ is the height of capitula and is set to 0.6 cm (Hájek
and Beckett, 2008).

**2.5 Model calibration for lateral water influence**
We used a machine-learning approach to estimate the influence of upstream area on the water
balance of the site. The rate of net inflow ($I$, see Eq. A18 in Appendix A.) was described as a
function of Julian day ($JD$), assuming the inflow was maximum after spring thawing and then
decreased linearly with time:
$$I_j = (a_N * JD + b_N) * Ks_j, JD > JD_{thaw} \tag{11}$$
where subscript $j$ denotes the peat layers under water table; $Ks$ is the saturated hydraulic
conductivity; $JD_{thaw}$ is the Julian day that thawing completed; and $a_N$ and $b_N$ are parameters.
We simulated water table changes using climatic scenarios from the Weather Generator
(Appendix A). During the calibration, the community compositions were set constant, such that
*S. magellanicum* fully occupied the hummock habitat whereas *S. fallax* fully occupied the lawn
habitat. The simulated multi-year means of weekly water table values were compared to the
weekly mean water table obtained observed at the site during years 2001, 2002, 2004 and 2016.
The cost function for the learning process was based on the sum of squared error (*SE*) of the
simulated water table:
$$SE = \Sigma(WTs_k - WTm_k)^2 \tag{12}$$
where *WTm* is the measured multi-year weekly mean of water table; *WTs* is the simulated multi-
year weekly mean of water table; and subscript $k$ denotes the week of year when the water table
was sampled.
The values of $a_N$ and $b_N$ were estimated using the Gradient Descent approach (Ruder, 2016),
by minimizing *SE* in above Eq. (19):
$$X_N(j) := X_N(j) - \Gamma \frac{\partial SE}{\partial X_N(j)} \tag{13}$$
where $\Gamma$ is the learning rate ($\Gamma = 0.1$). Appendix D shows the simulated water table with the
calibrated inflow term *I,* as compared against the measured values from the site.

**2.6 Model-based analysis**

First, we examined the ability of model to capture the preference of *S. magellanicum* for the
hummock environment and *S. fallax* for the lawn environment (Test 1). For both species, the
probability of occupation was initialized as 50% in a cell, and the distribution of species in the
communities were randomly patterned. Monte-Carlo simulations (40 replicates) were carried out,
with a time step of 30 minutes. A simulation length of 15 years was selected based on
preliminary studies, in order to cover the major interval of change and to ease computational
demand. Biomass growth, stem elongation and the spreading of shoots were simulated on a daily
basis. The establishment of new shoots in deactivated cells was calculated at the end of each
simulation year. We then assessed if the model could capture the dominance of *S. magellanicum*
in the hummock communities and the dominance of *S. fallax* in lawn communities. The
simulated annual height increments of mosses were compared to the values measured for each
community type. To measure moss height growth in the field, we deployed 20 cranked wires on
*S. magellanicum* dominated hummocks and 15 on *S. fallax* dominated lawns in 2016. Each
cranked wire was a piece of metal wire attached with plastic brushes at the side anchored into the
moss carpet (e.g. Clymo 1970, Holmgren et al., 2015). Annual height growth ($dH$) was
determined by measuring the change in the exposed wire length above moss surface from the
beginning to the end of growing season.
Second, we tested the robustness of the model to the uncertainties in a set of parameters (Test
2-4). In test 2, we focused on parameters that closely linked to hydrology and growth
calculations, but were roughly parameterized (e.g., $k_{imm}$, $r_{aero}$) or adopted as a prior from other
studies (e.g., $K_{sat}$, $\alpha$, $n$, $NSC_{max}$; see Table 3). One at a time, each parameter value was adjusted
by +10 % or -10. 40 Monte-Carlo simulations were run using the same runtime settings as in
Test 1. The simulated means of cover were then compared to those calculated without the
parameter adjustment.
Tests 3-4 were then carried out to test whether the model could correctly predict
competitiveness of the species in dry and wet habitats, if the species-specific trends of capitulum
water content were not correctly specified. For both species, we set the values of parameters
controlling the water retention (i.e. $B_{cap}$ and $D_S$, Appendix B) and the water-stress effects on net
photosynthesis (i.e. $W_{cap}$, Eq. 4) to be the same as those in *S. magellanicum* (Test 3) or same as
those in *S. fallax* (Test 4). Our hypothesis would be supported if removing the interspecific
differences in $RWC$ responses led to the failure to predict the habitat preferences of the species.
We implemented Tests 5-6 to test the importance of parameters that directly control the species
ability to overgrow another species with more rapid height increment (i.e. $Pm_{20}$, $Rs_{20}$, $\alpha_{PPFD}$ and
$H_{spec}$) in lawn and hummock conditions. We eliminated the species differences in the parameter
values to be same as those in *S. magellanicum* (Test 5) and same as those in *S. fallax* (Test 6).
The effects of the manipulation were compared against those from Tests 3-4. For each of Tests
3-6, 80 Monte-Carlo simulations were run using the setups described in Test 1.
Test 7-8 were implemented to separate the effects of photosynthetic water-response
parameters from the effects of the water retention of capitula. We set the photosynthetic water-
response parameters to be the same as those in *S. magellanicum* (Test 7) and same as those in *S.*
*fallax* (Test 8). As our model aimed to couple the environmental fluctuations and stochasticity of
ecosystem processes, we further tested the model responses to the absences of environmental
fluctuations (Test 9) or the absence of stochasticity in model parameters (Test 10). In Test 9,
monthly mean values of meteorological variables were used to drive the model simulation. In
Test 10, we removed the stochasticity of model parameters, and assigned average value to each
parameter of grid cells. For each of Tests 7-10, 40 Monte-Carlo simulations were run using the
setups described in Test 1.

**3 Results**
**3.1 Simulating the habitat preferences of *Sphagnum* species as affected by water content**
**traits of capitulum**
Test 1 demonstrated the ability of model to capture the preference of *S. magellanicum* for the
hummock environment and *S. fallax* for the lawn environment (Fig. 2A). The simulated annual
changes in species covers were greater in lawn than in hummock habitats during the first 5
simulation years. The changes in lawn habitats slowed down around year 10 and the cover of *S.*
*fallax* plateaued at around 95±2.8% (mean ±standard error). In contrast, the cover of *S.*
*magellanicum* on hummocks continued to grow until the end of simulation and reached
83±3.1%. In the lawn habitats, the cover of *S. fallax* increased in all Monte-Carlo simulations
and the species occupied all grid cells in 70% of the simulations. In the hummock habitats, the
cover of *S. magellanicum* increased in 91% of Monte-Carlo simulations, and formed
monocultural community in 16% of simulations (Fig. 2B). The height growth of *Sphagnum*
mosses was significantly greater at lawns than at hummocks (P<0.01). The ranges of simulated
height growths agreed well with the observed values from field measurement for both species
(Fig. 2C).

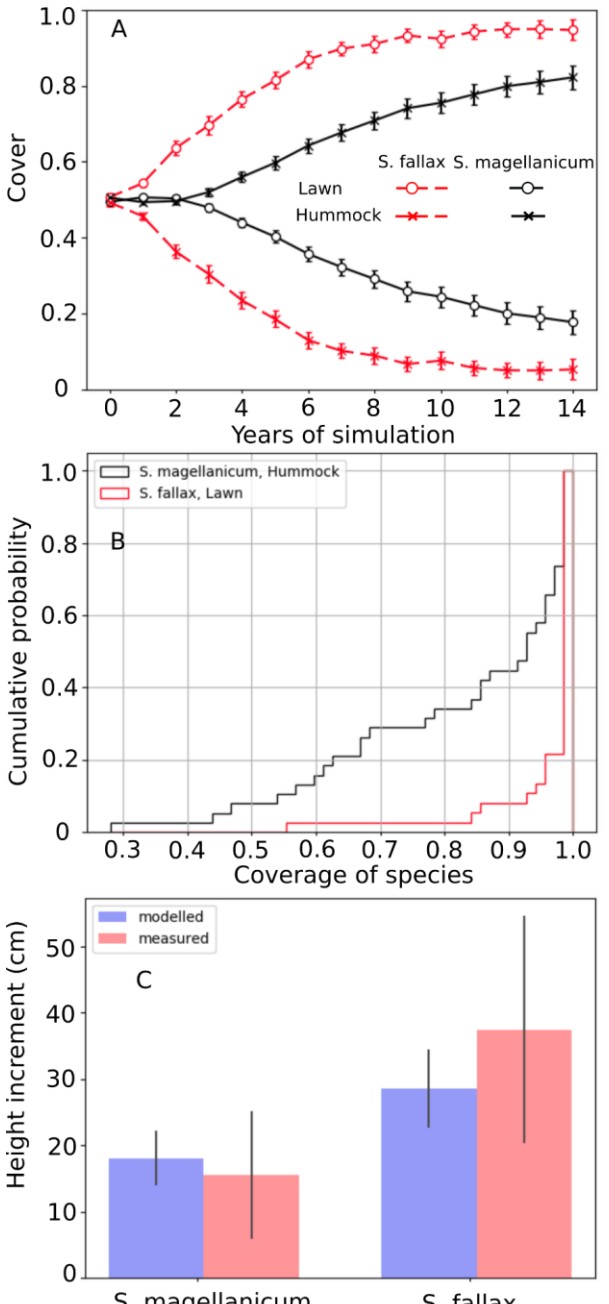


Figure 2. Testing the ability of PMS to predict habitat preference of *Sphagnum magellanicum* and *S. fallax* (Test 1). The hummock and lawn habitats were differentiated by water table depth, surface energy balances and capitulum water potential in modelling. In the beginning of simulation, the cover of the two species was set equal and it was allowed to develop with time. (A) Annual development of the relative cover (mean and standard error) of the two species in hummock and lawn habitats, (B) the cumulative probability distribution of the cover of the two species at the end of the 15-year period based on 40 Monte-Carlo simulations, and (C) the simulated and measured means of annual height growth of *Sphagnum* surfaces in their natural habitats in hummock and lawn habitats.


## 3.2 Testing model robustness


Test 2 addressed the model robustness to the uncertainties in several parameters that closely
linked to hydrology and growth calculations. Modifying most of the parameter values by +10%
or -10% yielded marginal changes in the mean cover of species in either hummock or hollow
habitat (Table 4). Reducing the moss carpet and peat hydraulic parameter *n* had stronger impacts
on *S. fallax* cover in hummocks than in lawns. Nevertheless, changes in simulated cover that
were caused by parameter manipulations were generally smaller than the standard deviations of
the means i.e. fitting into the random variation.

## 3.3 Testing the controlling role of capitulum water content for community dynamics


In Tests 3 and 4, the model incorrectly predicted the competitiveness of two species when the
interspecific differences of capitulum water content were eliminated. In both tests, *S. fallax*
became dominant in all habitats. The use of water responses characteristic to *S. magellanicum* for
both species (Test 3) led to faster development of *S. fallax* cover and higher coverage at the end
of simulation (Fig. 3A), as compared with the simulation results where the water responses
characteristic to *S. fallax* were used for both species (Test 4, Fig. 3B). The pattern was more
pronounced in hummock than in lawn habitats.

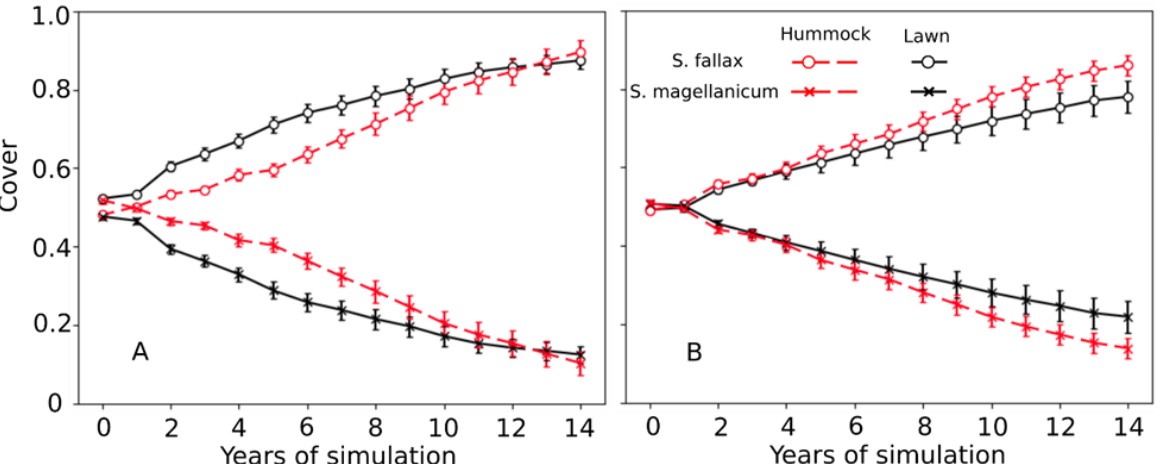


Figure 3. Testing the importance of capitulum water content to the habitat preference of *S.*
*magellanicum* and *S. fallax*. The development of the relative cover (mean and standard error)
were simulated in hummock and lawn habitats over a 15-year time frame for the two species. For
both species, parameter values for the capitulum water content, capitulum biomass ($B_{cap}$) and
density ($D_S$) were set to be the same as those from (A) *S. magellanicum* (Test 3) or (B) *S. fallax*
(Test 4).

In Tests 5 and 6, the species differences in the growth-related parameters were eliminated. However, the model still predicted the dominances of *S. fallax* and *S. magellanicum* in lawn and hummock habitats, respectively (Fig. 4). The increase in the mean cover of *S. magellanicum* was especially fast in hummock habitat in comparison to the results of the unchanged model from Test 1 (Fig. 2A). In lawns, the use of *S. fallax* growth parameters for both species gave stronger competitiveness to *S. magellanicum* (Fig. 4B) than using the *S. magellanicum* parameters (Fig. 4A). In Test 7 and 8, ignoring the interspecific differences in the photosynthetic water-response parameters did not change the simulated habitat preferences of *S. fallax* and *S. magellanicum* (Table 5). Using the water response parameters of *S. fallax* decreased the mean cover of *S. fallax* in lawns but increased the cover of *S. magellanicum* on hummocks. In contrast, using the water response parameters of *S. magellanicum* increased the mean cover of *S. fallax* in lawns but decreased the cover of *S. magellanicum* on hummocks.

432

433

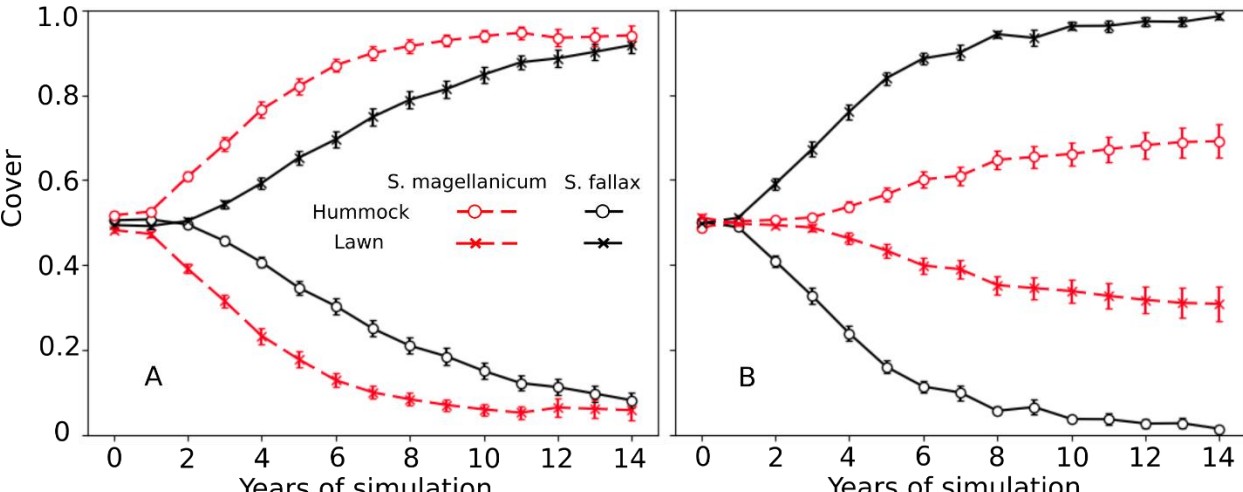

434

Figure 4. Testing the importance of parameters regulating net photosynthesis and shoot elongation to the habitat preference of *S. magellanicum* and *S. fallax*. Annual development of the relative cover (mean and standard error) of the two species were simulated for hummock and lawn habitats over a 15-year time frame. For both species, the parameter values (i.e. $Pm_{20}$, $Rs_{20}$, $\alpha_{PPFD}$ and $H_{spec}$) were set to be the same as those from (A) *S. magellanicum* (Test 5) or (B) *S. fallax* (Test 6).


**3.4 Testing the effects of environmental fluctuations and stochasticity of ecosystem processes on community dynamics**

In Tests 9, the model failed to simulate the preference of *S. magellanicum* to hummocks (Table
5) if the environmental fluctuation was ignored. However, the simulated cover of *S. fallax* in
lawns was higher as compared to unchanged condition (i.e. Test 1). Using mean value for each
model parameters led to mono output, i.e. *S. magellanicum* occupied 100% hummock area
whereas *S. fallax* took over lawns completely.


**4 Discussion**
In peatland ecosystems, *Sphagna* are keystone species distributed primarily along the
hydrological gradient (e.g. Andrus et al. 1986; Rydin, 1986). In a context where substantial
change in peatland hydrology is expected under a changing climate in northern areas (e.g. longer
snow-free season, lower summer water table and more frequent droughts), there is a pressing
need to understand how peatland plant communities could react and how *Sphagnum* species
could redistribute under habitat changes. In this work, we developed the Peatland Moss
Simulator (PMS), a process-based stochastic model, to simulate the competition between *S.*
*magellanicum* and *S. fallax*, two key species representing dry (hummock) and wet (lawn)
habitats in a poor fen peatland. We empirically showed that these two species differed in
characteristics that likely affect their competitiveness along a moisture gradient.
Capitulum water retention for the lawn-preferring species (*S. fallax*) was weaker than that for the
hummock-preferring species (*S. magellanicum*). Compared to *S. magellanicum*, the capitula of *S.*
*fallax* held less water at saturation and water content decreased more rapidly with dropping water
potential. Hence, *S. fallax* would dry faster than *S. magellanicum* under the same rate of water
loss. Moreover, water content in *S. fallax* capitula was less resistant to evaporation. These
differences indicated that it is harder for *S. fallax* capitula to buffer evaporative water loss and
thereby avoid or delay desiccation. Similar differences between hummock and hollow species
have been found earlier (Titus & Wagner, 1984; Rydin & McDonald, 1985). In addition, the net
photosynthesis of *S. fallax* is more sensitive to changes in capitulum water content than *S.*
*magellanicum,* as seen in a steeper decline in photosynthesis with decreasing water content (Fig.
B2C). Consequently, the growth of *S. fallax* is more likely to be slowed down by dry periods,
when the capillary water cannot fully compensate the evaporative loss (Robroek et al., 2007b)
making it less competitive in habitats prone to desiccation.
The PMS successfully captured the habitat preferences of the two *Sphagnum* species (Test 1):
starting from a mixed community with equal probabilities for both species, the lawn habitats
with shallower water table were eventually dominated by the typical lawn species *S. fallax*,
whereas hummock habitats, which were 15 cm higher than the lawn surface, were taken over by
*S. magellanicum*. The low final cover of *S. magellanicum* simulated in lawn habitats agreed well
with field observation from our study site, where *S. magellanicum* cover was less than 1% in
lawns (Kokkonen et al., 2019). On the other hand, *S. fallax* was outcompeted by *S. magellanicum*
in the hummock habitats. This result is consistent with previous findings that hollow-preferring
*Sphagna* are less likely to survive in hummock environments with greater drought pressure (see
Rydin 1985; Rydin et al. 2006; Johnson et al., 2015). The simulated annual height increments of
mosses also agreed well with the observed values for both habitat types. Our simulation for lawn
habitat shows that the looser stem structure of *S. fallax* allows it to allocate more of its produced
biomass into height growth, and thereby overgrow *S. magellanicum,* in which new biomass
forms a compact stem, packed with thick fascicles. This finding indicates that PMS can capture
key mechanisms in controlling the growth and competitive interactions of the *Sphagnum* species.
Parameter sensitivity testing showed the robustness of PMS regarding the uncertainties in
parameterization, as the simulated changes in the mean species cover, under 10% changes in
several parameters, were generally less than the standard deviations of the means. Decreasing the
value of the hydraulic parameter *n* (Table 3, Eq. A5) increased the presence of *S. fallax* in the
hummock habitats. This was expected as *n* is a scaling factor and therefore its changes get
magnified: a lower *n* value will lead to higher water content in the unsaturated layers above the
water table (van Genuchten, 1978), which allows wet-adapted *Sphagna* to survive dry conditions
(Hayward and Clymo, 1982; Robroek et al., 2007b; Rice et al., 2008). In contrast, the response
of *Sphagnum* cover to the changes in other hydraulic parameters (i.e. $\alpha$, *n*, $K_h$) was limited in
lawn habitats. This could be due to the relatively shallow water table in lawns, which was able to
maintain sufficient capillary rise to the moss carpet and capitula. Decreasing the values of the
specific immobilization rate (*kimm)* and maximal NSC concentration in *Sphagnum* biomass
*(NSC_{max})* mainly decreased the cover of *S. fallax* in lawn habitats, consistent with the importance
of biomass production to *Sphagna* in high moisture environment (e.g. Rice et al., 2008; Laine et
al., 2011).  In addition, the SVAT modelling for hummocks and lawns (Module III, Fig. 1)
employed same hydraulic parameter values obtained from *S. magellanicum* hummocks
(McCarter and Price, 2014). For lawns, this could overestimate $K_m$ but underestimate *n*, as the
lawn peat would be' be less efficient in holding high water content and generating capillary-flow
than hummock peat (Robroek et al., 2007b; Branham and Strack, 2014).  As the decrease in $K_m$
and increase in *n* showed counteracting effects on the simulated species covers (Table 4), the
biases in the parameterization of $K_m$ and *n* may not critically impact model performance.
Both our empirical measurements and PMS simulations indicate the importance of capitulum
water content as a mechanism controlling the moss community dynamics in peatlands. It has
long been hypothesized and experimentally studied that *Sphagnum* niche is defined by two
processes. Firstly, dry, high elevation habitats such as hummocks physically select species with
ability to remain moist (Rydin, 1993). If the interspecific differences in water retention and
water-stress effects were correctly specified (Test 1, Fig. 2) our model predicted this phenomena
of stronger competitiveness of *S. magellanicum* against *S. fallax* in hummock habitats correctly.
Alternatively, the model failed to predict the distribution of *S. magellanicum* on hummocks, if
these interspecific differences in the water processes were neglected (Test 3 and Test 4, Fig. 3).
During low water table periods in summer capillary rise may not fully compensate for high
evaporation (Robroek et al., 2007b; Nijp et al., 2014). In such circumstances, capitulum water
potential could drop rapidly towards the pressure defined by the relative humidity of air
(Hayward and Clymo, 1982). Consequently, the ability of capitula to retain cytoplasmic water is
particularly important for the hummock-preferring species, as was also shown by Titus &
Wagner (1984).
Secondly, in habitats with more persistently high moisture content such as lawns and hollows,
interspecific competition becomes important: it is well acknowledged that species from such
habitats generally have higher growth rates and photosynthetic capacity compared to hummock
species (e.g. Laing et al., 2014; Bengtsson et al., 2016). Our results also agreed on this, as setting
the growth-related parameters (i.e. $Pm_{20}$, $Rs_{20}$, $\alpha_{PPFD}$ and $H_{spec}$) of *S. magellanicum* to be the
same as those of *S. fallax* decreased the *S. fallax* cover in both hummock and lawn habitats (Test
6, Fig. 4B). However, such changes didn't impact the simulated habitat preferences for the tested
species. Based on this, the growth-related parameters seem to be less important than those water-
related ones. Further on, our Tests 7 and 8 showed that when interspecific differences in the
water-stress effects on photosynthesis were removed, the model still predicted the correct habitat
preferences of *S. magellanicum* and *S. fallax*. Therefore, the interspecific differences in
capitulum water retention could be the main determinant on the habitat preferences of the tested
species.

There have been growing concerns about the shift of peatland communities from *Sphagnum*-
dominated towards more vascular-abundant under a drier and warmer climate (Wullschleger et
al., 2014; Munir et al. 2015; Dieleman et al. 2015). Nevertheless, the potential of *Sphagnum*
species composition to adjust to this forcing remains poorly understood. Particularly in
oligotrophic fens, where the vegetation is substantially shaped by lateral hydrology
(Tahvanainen, 2011; Turetsky et al., 2012), plant communities can be highly vulnerable to
hydrological changes (Gunnarsson et al. 2002; Tahvanainen, 2011). Based on the validity and
robustness of PMS, we believe PMS could serve as one of the first mechanistic tools to
investigate the direction and rate of change in *Sphagnum* communities under environmental
forcing. The hummock-lawn differences showed by Test 1 imply that *S. magellanicum* could
outcompete *S. fallax* within a decade in a poor fen community, if the water table of habitats like
lawns was lowered by 15 cm (Test 1). Although this was derived from a simplified system with
only the two species, it highlighted the potential of rapid turnover of *Sphagnum* species: the
hummock-lawn difference of water table in simulation was comparable to the expected water
table drawdown in fens under the warming climate (Whittington and Price, 2006; Gong et al.,
2013b). The effect traits of mosses, while studied less than those of vascular plant traits, have far
reaching impacts on biogeochemistry of ecosystems such as peatlands, where mosses form the
most significant plant group (Cornelissen et al. 2007). Because of the large interspecific
differences of traits such as photosynthetic potential, hydraulic properties and litter chemistry
(Laiho 2006; Straková et al., 2011; Korrensalo et al., 2017; Jassey & Signarbieux, 2019), change
in *Sphagnum* community composition is likely to impact long-term peatland stability and
functioning (Waddington et al., 2015). Turnover between hummock and wetter habitat species
would feedback to climate as they differ in their decomposability (Straková et al. 2012;
Bengtsson et al. 2016). As hummock species produces more calcitrant litter the carbon bind into
the system would take longer to get released back to atmosphere. In addition, the replacement of
wet adapted moss species with hummock species is likely to result in higher ability to maintain
carbon sink under periods of drought (Jassey, & Signarbieux, 2019).
Although efforts have been made on analytical modelling to obtain boundary conditions for
equilibrium states of moss and vascular communities in peatland ecosystems (Pastor et al.,
2002), the dynamic process of peatland vegetation has not been well-described or included in
earth system models (ESMs). Existing ecosystem models usually consider the features of
peatland moss cover as "fixed" (Sato et al., 2007; Wania et al., 2009; Euskirchen et al., 2014), or
change directionally following a projected trajectory (Wu and Roulet, 2014). Chaudhury et al.
(2017) have a dynamic peatland vegetation model, with a single moss PFT and four vascular
PFTs, so moss productivity relative to vascular plants can vary, however moss characteristics are
fixed to a single set of values. Our modelling approach provided a way to incorporate the
environmental fluctuation and the mechanisms of dynamic moss cover into peatland carbon
modelling. PMS employed an individual-based approach where each grid cell carries a unique
set of trait properties, so that shoots with favorable trait combinations in prevailing environment
are thus able to replace those whose trait combinations are less favorable. Moreover, the model
included the spatial interactions of individuals, which can impact the sensitivity of coexistence
pattern to environmental changes (Bolker et al., 2003; Sato et al., 2007; Tatsumi et al., 2019).
This mimics the stochasticity in plant responses to environmental fluctuations, which is essential
to community assembly and trait filtering under environmental forcing (Clark et al., 2010). The
importance of incorporating environmental fluctuations with the stochasticity of biophysiological
processes is supported by our Test 9 and 10. If the monthly mean climate conditions were used
as input, our model failed to predict the dominance of *S. magellanicum* on hummocks. If the
stochasticity of model parameters were omitted and only mean values were used, the model
generated only single output disregarding the randomness of environmental conditions. As these
features are considered essential to the "next generation" DVMs (Scheiter et al., 2013), our PMS
could be considered as an elemental design for future DVM development.
We see PMS as an elemental design for the future development of dynamic vegetation models
for peatland ecosystems, yet there are certain uncertainties and features that should be developed
further. We used a gas-exchange-based method to quantify the simultaneous changes in capitula
water potential, water content and carbon uptake of *Sphagnum* moss capitula. It should be noted
that, the measurements mainly covered the changes from $RWC_{opt}$ towards $RWC_{cmp}$ (Table 1 and
Fig. 3). However, capitula water content could be higher than $RWC_{opt}$ at saturation (e.g. about
25-30 g g$^{-1}$; Schipperges and Rydin, 1998). When RWC is high, vapor diffusion may occur
mainly from the capitula surface or macropores, instead of the inside capitula. Hence, our
methodology may not be suitable to reflect the water potential changes under near-saturation
conditions. In our modelling, we used the volumetric water content of moss carpet to estimate
RWC as an approximation for wet conditions (Eq. 17). The accuracy of such approximation for
high RWC conditions remains ambiguous and more information is still required.
We assumed that tissue structure did not change during the measurement process, and that the
aerodynamic resistance ($r_a$, Eq. 3) for vapor to diffuse from the inner capitula to the headspace
was constant. However, capitula drying may change leaf curvature, especially in species with
slim and sparsely spread leaves (Laine et al., 2018). Such changes in the branch-leaf structure
could expose the more of the leaf surface to evaporation and reduce the value of $r_a$.
Consequently, PMS could underestimate capitula water potential towards the drying end for
those species, if a constant $r_a$ is derived from the maximal evaporation rate ($E_m$, Eq. 5; Fig 3C).
The water-retention relationship in PCM may not sufficiently capture water potential changes at
wet and dry extremes (e.g., *S. magellanicum* in Fig. 4C). Water retention functions developed
for mineral soils (e.g., Clapp and Hornberge, 1978; van Genuchten, 1980) may not be well
parameterized for peat soils and moss (non-vacular) vegetation, particularly under very dry or
wet conditions. Hence, further studies are needed to improve the description of the nonlinearity
of capitula water content, as influenced by capitula morphology (e.g. capitula biomass and shoot
density) and structural changes of branch leaves.
PMC lacks horizontal (lateral) water transport that may allow individuals of lawn species to be
present in hummocks (Rydin 1985). With additional experimental data, such as species-specific
hydraulic conductivity, the current model could be improved to also quantify the horizontal
water transport among neighboring grid cells.
To conclude, PMS could successfully capture the habitat preferences of the modelled *Sphagnum*
species. In this respect, PMS could provide fundamental support for the future development of
dynamic vegetation models for peatland ecosystems. Based on our findings, capitulum water
processes should be considered as a control on vegetation dynamics in future impact studies on
peatlands under changing environmental conditions.

**Acknowledgements**
We are grateful to Harri Strandman (University of Eastern Finland) for the coding of Weather
Generator. The project was funded by Academy of Finland (Project number 287039). AML
acknowledges support from the Kone Foundation and SF from grant #1802825 from the US
National Science Foundation, and the Fulbright-Finland and Saastamoinen Foundations.

*Code and data availability.* The data and the code to reproduce the analysis is available upon
request to the corresponding author.
*Author contributions.* JG and EST designed the study. JG, AML and NK conducted the
experiment and analysis. JG, EST, NR and SF designed the model. JG coded the model and
conducted the model simulation and data analysis. JG and EST wrote the manuscript with
contributions from all co-authors.
*Competing interests.* The authors declare that they have no conflict of interest.

**References**
Alm, J., Shurpali, N. J., Tuittila, E.-S., Laurila, T., Maljanen, M., Saarnio, S. and Minkkinen, K.:
Methods for determining emission factors for the use of peat and peatlands – flux measurements
and modelling, Boreal Environment Research, 12, 85-100, 2007.
Amarasekare, P.: Competitive coexistence in spatially structured environments: A synthesis,
Ecology Letters, 6, 1109-1122, 2003.
Anderson K. and Neuhauser C.: Patterns in spatial simulations—are they real? Ecological
Modelling, 155, 19-30, 2000.
Andrus R. E.: Some aspects of Sphagnum ecology, Can. J. Bot., 64, 416–426, 1986.
Asaeda, T. and Karunaratne, S.: Dynamic modelling of the growth of Phragmites australis:
model description, Aquatic Botany, 67, 301-318, 2000.
Bengtsson, F., Granath, G. and Rydin, H.: Photosynthesis, growth, and decay traits in Sphagnum
- a multispecies comparison. Ecology and Evolution, 6, 3325-3341, 2016.
Blois, J. L., Williams, J. W., Fitzpatrick, M. C., Jackson, S. T. and Ferrier S.: Space can
substitute for time in predicting climate-change effects on biodiversity, PNAS, 110, 9374-9379,
doi:10.1073/pnas.1220228110, 2013.
Breeuwer, A., Heijmans, M. M., Robroek, B. J. and Berendse, F.: The effect of temperature on
growth and competition between Sphagnum species. Oecologia, 156(1), 155-67, 2008.
Bolker, B. M., Pacala, S. W. and Neuhauser, C.: Spatial dynamics in model plant communities:
What do we really know? Am. Nat., 162, 135–148, 2003.

Boulangeat, I., Svenning, J. C., Daufresne, T., Leblond, M. and Gravel, D.: The transient response of ecosystems to climate change is amplified by trophic interactions, Oikos, 127, 1822–1833, 2018.

Branham, J. E. and Strack, M.: Saturated hydraulic conductivity in *Sphagnum*-dominated peatlands: do microforms matter? Hydrol. Process., 28, 4352-4362, 2014.

Chaudhary, N., Miller, P. A., and Smith, B. Modelling Holocene peatland dynamics with an individual-based dynamic vegetation model, Biogeosciences, 14, 2571–2596, 2017.

Chesson, P.: General theory of competitive coexistence in spatially varying environments. Theoretical Population Biology 58, 211–237, 2000.

Clapp, R. B. and Hornberger, G. M.: Empirical equations for some soil hydraulic properties. Water Resour. Res, 14, 601–604, 1978.

Clark J. S., Bell D., Chu C., Courbaud B., Dietze M., Hersh M., HilleRisLambers J., Ibanez I., LaDeau S., McMahon S., Metcalf, J., Mohan, J., Moran, E., Pangle, L., Pearson, S., Salk, C., Shen, Z., Valle, D. and Wyckoff, P.: High-dimensional coexistence based on individual variation: a synthesis of evidence, Ecological Monographs, 80, 569 – 608, 2010.

Clymo, R. S.: The growth of Sphagnum: Methods of measurement, Journal of Ecology, 58, 13-49, 1970.

Cornelissen, J. H., Lang, S. I., Soudzilovskaia, N. A., and During, H. J.: Comparative cryptogam ecology: a review of bryophyte and lichen traits that drive biogeochemistry. Annals of botany, 99(5), 987-1001, 2007

Czárán T. and Iwasa Y.: Spatiotemporal models of population and community dynamics, Trends Ecol. Evol., 13, 294–295, 1998.

Dieleman, C. M., Branfireun, B. A., Mclaughlin, J. W. and Lindo, Z.: Climate change drives a shift in peatland ecosystem plant community: Implications for ecosystem function and stability, Global Change Biology, 21, 388-395, 2015.

Euskirchen, E. S., Edgar, C. W., Turetsky, M. R., Waldrop, M. P. and Harden J. W.: Differential response of carbon fluxes to climate in three peatland ecosystems that vary in the presence and stability of permafrost, permafrost, J. Geophys. Res. Biogeosci., 119, 1576–1595, 2014.

Frolking, S., Roulet, N. T., Moore, T. R., Lafleur, T. M., Bubier, L. J. and Crill, P. M.: Modeling seasonal to annual carbon balance of Mer Bleue Bog, Ontario, Canada. Global Biogeochem. Cycles, 16, doi:10.1029/2001GB001457, 2002.

Gassmann, F., Klötzli, F. and Walther, G.: Simulation of observed types of dynamics of plants and plant communities, Journal of Vegetation Science, 11, 397 – 408, 2003.

Goetz, J. D. and Price, J. S.: Role of morphological structure and layering of *Sphagnum* and
*Tomenthypnum* mosses on moss productivity and evaporation rates. Canadian Journal of Soil
Science, 95, 109-124, 2015.
Gong, J., Shurpali, N., Kellomäki, S., Wang, K., Salam, M. M. and Martikainen, P. J.: High
sensitivity of peat moisture content to seasonal climate in a cutaway peatlandcultivated with a
perennial crop (Phalaris arundinacea, L.): a modeling study, Agricultural and Forest
Meteorology, 180, 225–235, 2013a.
Gong, J., Wang, K., Kellomäki, S., Wang, K., Zhang, C., Martikainen, P. J. and Shurpali, N.:
Modeling water table changes in boreal peatlands of Finland under changing climate conditions,
Ecological Modelling, 244, 65-78, 2013b.
Gong, J., Jia, X., Zha, T., Wang, B., Kellomäki, S. and Peltola, H.: Modeling the effects of plant-
interspace heterogeneity on water-energy balances in a semiarid ecosystem, Agricultural and
Forest Meteorology, 221, 189–206, 2016.
Gorham, E.: Northern peatlands: Role in the carbon cycle and probable responses to climatic
warming, Ecol. Appl., 1, 182–195, 1991.
Gunnarsson, U., Malmer, N. and Rydin, H.: Dynamics or constancy in Sphagnum dominated
mire ecosystems? A 40-year study, Ecography, 25, 685–704, 2002.
Hartmann, H. and Trumbore, S.: Understanding the roles of nonstructural carbohydrates in forest
trees – from what we can measure to what we want to know, New Phytol, 211, 386-403, 2016.
Hájek, T. and Beckett, R. P.: Effect of water content components on desiccation and recovery in
Sphagnum mosses, Annals of Botany, 101, 165–173, 2008.
Hájek, T., Tuittila, E.-S., Ilomets, M. and Laiho, R.: Light responses of mire mosses - A key to
survival after water-level drawdown? Oikos, 118, 240-250, 2009.
Hayward P. M. and Clymo R. S.: Profiles of water content and pore size in Sphagnum and peat,
and their relation to peat bog ecology. Proceedings of the Royal Society of London, Series B,
Biological Sciences, 215, 299-325, 1982.
Hayward P. M. and Clymo R. S.: The growth of Sphagnum: experiments on, and simulation of,
some effects of light flux and water-table depth. Journal of Ecology, 71, 845-863, 1983.
Holmgren, M., Lin, C., Murillo, J. E., Nieuwenhuis, A., Penninkhof, J., Sanders, N., Bart, T.,
Veen, H., Vasander, H., Vollebregt, M. E. and Limpens, J.: Positive shrub–tree interactions
facilitate woody encroachment in boreal peatlands, J. Ecol., 103, 58-66, 2015.
Hugelius, G., Tarnocai, C., Broll, G., Canadell, J. G., Kuhry, P. and Swanson, D. K.: The
Northern Circumpolar Soil Carbon Database: spatially distributed datasets of soil coverage and
soil carbon storage in the northern permafrost regions, Earth Syst. Sci. Data, 5, 3-13, 2013.
Jassey, V. E., & Signarbieux, C.: Effects of climate warming on Sphagnum photosynthesis in
peatlands depend on peat moisture and species-specific anatomical traits. *Global change biology*,
*25*(11), 3859-3870, 2019.
Johnson, M. G., Granath, G., Tahvanainen, T., Pouliot, R., Stenøien, H. K., Rochefort, L., Rydin,
H. and Shaw, A. J.: Evolution of niche preference in Sphagnum peat mosses, Evolution, 69, 90 –
734  103, 2015.

Kellomäki, S. and Väisänen, H.: Modelling the dynamics of the forest ecosystem for climate
change studies in the boreal conditions, Ecol. Model., 97, 121-140, 1997.
Keuper, F., Dorrepaal, E., Van Bodegom, P. M., Aerts, R., Van Logtestijn, R. S.P., Callaghan, T.
V. and Cornelissen, J. H.C.: A Race for Space? How Sphagnum fuscum stabilizes vegetation
composition during long-term climate manipulations, Global Change Biology, 17, 2162–2171,
740  2011.

Kokkonen, N., Laine, A., Laine, J., Vasander, H., Kurki, K., Gong, J. and Tuittila, E.-S.:
Responses of peatland vegetation to 15-year water level drawdown as mediated by fertility level.
J. Veg. Sci., 30(6), 1206-1216, 2019.
Korrensalo, A., Hájek, T., Vesala, T., Mehtätalo, L., and Tuittila, E. S.: Variation in
photosynthetic properties among bog plants. Botany, 94(12), 1127-1139, 2016.
Korrensalo, A., Alekseychik, P., Hájek, T., Rinne, J., Vesala, T., Mehtätalo, L., Mammarella, I.
and Tuittila, E.-S.: Species-specific temporal variation in photosynthesis as a moderator of
peatland carbon sequestration, Biogeosciences, 14, 257-269, 2017.
Kyrkjeeide, M. O., Hassel, K., Flatberg, K. I., Shaw, A. J., Yousefi, N. and Stenøien, H. K.
Spatial genetic structure of the abundant and widespread peatmoss Sphagnum magellanicum
Brid. PLoS One, 11, e0148447, 2016.
Laiho, R. Decomposition in peatlands: Reconciling seemingly contrasting results on the impacts
of lowered water levels, Soil Biology and Biochemistry, 38, 2011-2024, 2006.
Laine, A. M. Juurola, E., Hájek, T., and Tuittila, E.-S.: Sphagnum growth and ecophysiology
during mire succession. Oecologia, 167: 1115-1125, 2011.
Laine, J., Komulainen, V.-M., Laiho, R., Minkkinen, K., Ras- inm̈aki, A., Sallantaus, T.,
Sarkkola, S., Silvan, N., Tolonen, K., Tuittila, E.-S., Vasander, H., and Päivänen, J.:. Lakkasuo –
a guide to mire ecosystem, Department of Forest Ecology Publications, University of Helsinki,
31, 123 pp, 2004.
Laine, J., Flatberg, K. I., Harju, P., Timonen, T., Minkkinen, K., Laine, A., Tuittila, E.-S. and
Vasander, H.: Sphagnum Mosses — The Stars of European Mires. University of Helsinki
Department of Forest Sciences, Sphagna Ky. 326 p, 2018
Laine J., Harju P., Timonen T., Laine A., Tuittila E.-S., Minkkinen K. and Vasander H.: The
inticate beauty of Sphagnum mosses—a Finnish guide to identification (Univ Helsinki Dept
Forest Ecol Publ 39). Department of Forest Ecology, University of Helsinki, Helsinki, pp 1–190,
766 2009.

Laing, C. G., Granath, G., Belyea, L. R., Allton K. E. and Rydin, H.: Tradeoffs and scaling of
functional traits in Sphagnum as drivers of carbon cycling in peatlands, Oikos, 123, 817–828,
769 2014.

Larcher, W.: Physiological Plant Ecology: Ecophysiology and Stress Physiology of Functional
Groups, Springer, 2003.
Letts, M. G., Roulet, N. T. and Comer, N. T.: Parametrization of peatland hydraulic properties
for the Canadian land surface scheme, Atmosphere-Ocean, 38, 141-160, 2000.
Martínez-Vilalta, J., Sala, A., Asensio, D., Galiano, L., Hoch, G., Palacio, S., Piper, F. I. and
Lloret, F.: Dynamics of non-structural carbohydrates in terrestrial plants: a global synthesis. Ecol
Monogr, 86, 495-516, 2016.
McCarter C. P. R. and Price J. S.: Ecohydrology of Sphagnum moss hummocks: mechanisms of
capitula water supply and simulated effects of evaporation. Ecohydrology 7, 33 – 44, 2014.
Munir, T. M., Perkins, M., Kaing, E. and Strack, M.: Carbon dioxide flux and net primary
production of a boreal treed bog: Responses to warming and water-table-lowering simulations of
climate change, Biogeosciences, 12, 1091–1111, 2015.
Murray, K. J., Harley, P. C., Beyers, J., Walz, H. and Tenhunen, J. D.: Water content effects on
photosynthetic response of Sphagnum mosses from the foothills of the Philip Smith Mountains,
Alaska, Oecologia, 79, 244-250, 1989.
Nijp, J. J., Limpens, J., Metselaar, K., van der Zee, S. E. A. T. M., Berendse, F. and Robroek B.
J. M.: Can frequent precipitation moderate the impact of drought on peatmoss carbon uptake in
northern peatlands? New Phytologist, 203, 70-80, 2014.
O'Neill, K. P.: Role of bryophyte-dominated ecosystems in the global carbon budget. In A. J.
Shaw and B. Goffi net [eds.] Bryophyte biology, 344–368, Cambridge University Press,
Cambridge, UK, 2000.
Pastor, J., Peckham, B., Bridgham, S., Weltzin, J. and Chen J.: Plant community dynamics,
nutrient cycling, and alternative stable equilibria in peatlands. American Naturalist, 160, 553-
793 568, 2002.

Päivänen, J.: Hydraulic conductivity and water retention in peat soils, Acta Forestalia Fennica,
795 129, 1-69, 1973.

Pouliot, R., Rochefort, L., Karofeld, E. and Mercier, C.: Initiation of Sphagnum moss hummocks
in bogs and the presence of vascular plants: Is there a link? Acta Oecologica, 37, 346-354, 2011.
Price, J. S., Whittington, P. N., Elrick, D. E., Strack, M., Brunet, N. and Faux, E.: A method to
determine unsaturated hydraulic conductivity in living and undecomposed moss, Soil Sci. Soc.
Am. J., 72, 487 – 491, 2008.
Price, J. S. and Whittington, P. N.: Water flow in Sphagnum hummocks: Mesocosm
measurements and modelling, Journal of Hydrology 381, 333 – 340, 2010.
Rice, S. K., Aclander, L. and Hanson, D. T.: Do bryophyte shoot systems function like vascular
plant leaves or canopies? Functional trait relationships in Sphagnum mosses (Sphagnaceae),
American Journal of Botany, 95, 1366-1374, 2008.
Riutta, T., Laine, J., Aurela, M., Rinne, J., Vesala, T., Laurila, T., Haapanala, S., Pihlatie, M. and
Tuittila, E.-S.: Spatial variation in plant community functions regulates carbon gas dynamics in a
boreal fen ecosystem, Tellus, 59B, 838-852, 2007.
Robroek, B. J.M., Limpens, J., Breeuwer, A., Crushell, P. H. and Schouten, M. G.C.:
Interspecific competition between Sphagnum mosses at different water tables, Functional
Ecology, 21, 805 – 812, 2007a.
Robroek, B. J.M., Limpens, J., Breeuwer, A., van Ruijven, J. and Schouten, M. G.C.:
Precipitation determines the persistence of hollow Sphagnum species on hummocks, Wetlands,
4, 979 – 986, 2007b.
Robroek, B. J.M., Schouten, M. G.C., Limpens, J., Berendse, F. and Poorter, H.: Interactive
effects of water table and precipitation on net $CO_2$ assimilation of three co-occurring Sphagnum
mosses differing in distribution above the water table, Global Change Biology 15, 680 – 691,
818 2009.

Ruder, S.: An overview of gradient descent optimization algorithms, CoRR, abs/1609.04747,
820 2016.

Runkle, B.R.K., Wille, C., Gažovič M., Wilmking, M. and Kutzbach, L.: The surface energy
balance and its drivers in a boreal peatland fen of northwestern Russia, Journal of Hydrology,
823 511, 359-373, 2014.

Rydin, H: Interspecific competition between Sphagnum mosses on a raised bog. Oikos, 413-423,
825 1993.

Rydin, H.: Competition and niche separation in Sphagnum. Canadian Journal of Botany, 64(8),
827 1817-1824, 1986.

Rydin, H.: Competition between Sphagnum species under controlled conditions. Bryologist, 302-
307, 1997.Rydin, H. and McDonald A. J. S.: Tolerance of Sphagnum to water level. Journal of
Bryology, 13, 571–578, 1985.
Rydin, H., Gunnarsson, U., and Sundberg, S.: The role of Sphagnum in peatland development

and persistence, in: Boreal peatland ecosystems, edited by: Wieder, R. K., and Vitt, D. H.,30 Ecological Studies Series, Springer Verlag, Berlin, 47–65, 2006.

Sato, H., Itoh, A. and Kohyama, T.: SEIB-DGVM: A new Dynamic Global Vegetation Model using a spatially explicit individual-based approach, Ecol. Model., 200, 279–307, 2007.

Scheiter, S., Langan, L. and Higgins, S. I.: Next-generation dynamic global vegetation models: learning from community ecology, New Phytologist, 198, 957-969, 2013.

Schipperges, B. and Rydin, H.: Response of photosynthesis of Sphagnum species from contrasting microhabitats to tissue water content and repeated desiccation, The New Phytologist, 140, 677-684, 1998.

Silvola, J., Aaltonen, H.: Water content and photo- synthesis in the peat mosses Sphagnum fuscum and S. angustifolium. Annales Botanici Fennici 21, 1–6, 1984.

Smirnoff, N.: The carbohydrates of bryophytes in relation to desiccation tolerance, Journal of Bryology, 17, 185-19, 1992.

Straková, P., Niemi, R. M., Freeman, C., Peltoniemi, K., Toberman, H., Heiskanen, I., Fritze, H. and Laiho, R.: Litter type affects the activity of aerobic decomposers in a boreal peatland more than site nutrient and water table regimes, Biogeosciences, 8, 2741-2755, 2011.

Straková, P., Penttilä, T., Laine, J., and Laiho, R.: Disentangling direct and indirect effects of water table drawdown on above-and belowground plant litter decomposition: consequences for accumulation of organic matter in boreal peatlands. Global Change Biology, 18, 322-335, 2012.

Strandman, H., Väisänen, H. and Kellomäki, S.: A procedure for generating synthetic weather records in conjunction of climatic scenario for modelling of ecological impacts of changing climate in boreal conditions, Ecol. Model., 70, 195–220, 1993.

Szurdoki, E., Márton, O., Szövényi, P.: Genetic and morphological diversity of *Sphagnum angustifolium*, *S. flexuosum* and *S. fallax* in Europe. Taxon, 63, 237–48, 2014.

Tahvanainen, T.: Abrupt ombrotrophication of a boreal aapa mire triggered by hydrological disturbance in the catchment, Journal of Ecology, 99, 404-415, 2011.

Tatsumi, S., Cadotte M. W. and Mori, A. S.: Individual-based models of community assembly: Neighbourhood competition drives phylogenetic community structure, J. Ecol., 107, 735–746, 2019.

Thompson, D. K., Baisley, A. S. and Waddington, J. M.: Seasonal variation in albedo and radiation exchange between a burned and unburned forested peatland: implications for peatland evaporation, Hydrological Processes, 29, 3227-3235, 2015.

Titus, J. E., and Wagner, D. J.: Carbon balance for two Sphagnum mosses: water balance resolves a physiological paradox. Ecology, 65(6), 1765-1774, 1984.

Turetsky, M. R.: The role of bryophytes in carbon and nitrogen cycling, Bryologist, 106, 395 – 409, 2003.

Turetsky, M. R., Crow, S. E., Evans, R. J., Vitt, D. H. and Wieder, R. K.: Trade-offs in resource allocation among moss species control decomposition in boreal peatlands, Journal of Ecology, 96, 1297-1305, 2008.

Turetsky, M. R., Bond-Lamberty, B., Euskirchen, E., Talbot, J., Frolking, S., McGuire, A. D. and Tuittila, E.: The resilience and functional role of moss in boreal and arctic ecosystems, New Phytologist, 196, 49-67, 2012.

van Gaalen, K. E., Flanagan, L. B., Peddle, D. R.: Photosynthesis, chlorophyll fluorescence and spectral reflectance in Sphagnum moss at varying water contents. Oecologia, 153, 19 – 28, 2007.

van Genuchten, M.: A closed-form equation for predicting the hydraulic conductivity of unsaturated soils, Soil Science Society of American Journal, 44, 892–898, 1980.

Väliranta, M., Korhola, A., Seppä, H., Tuittila, E. S., Sarmaja-Korjonen, K., Laine, J. and Alm, J.: High-resolution reconstruction of wetness dynamics in a southern boreal raised bog, Finland, during the late Holocene: a quantitative approach, The Holocene, 17, 1093–1107, 2007.

Venäläinen, A., Tuomenvirta, H., Lahtinen, R. and Heikinheimo, M.: The influence of climate warming on soil frost on snow-free surfaces in Finland, Climate Change, 50, 111-128, 2001.

Vionnet, V., Brun, E., Morin, S., Boone, A., Faroux, S., Le Moigne, P., Martin, E. and Willemet, J.-M.: The detailed snowpack scheme Crocus and its implementation in SURFEX v7.2, Geoscientific Model Development, 5, 773-791, 2012

Vitt, D. H.: Peatlands: Ecosystems dominated by bryophytes. In A. J. Shaw and B. Goffi net [eds.], Bryophyte biology, 312 – 343, Cambridge University Press, Cambridge, UK, 2000.

Waddington, J. M., Morris, P. J., Kettridge, N., Granath, G., Thompson, D. K. and Moore, P. A.: Hydrological feedbacks in northern peatlands, Ecohydrology, 8, 113 – 127, 2015.

Wania, R., Ross, I. and Prentice, I. C.: Integrating peatlands and permafrost into a dynamic global vegetation model: 2. Evaluation and sensitivity of vegetation and carbon cycle processes, Global Biogeochemical Cycles, 23, GB3015, DOI:10.1029/2008GB003413, 2009.

Weiss, R., Alm, J., Laiho, R. and Laine, J.: Modeling moisture retention in peat soils, Soil Science Society of America Journal, 62, 305–313, 1998.

Whittington, P. N. and Price, J. S.: The effects of water table draw-down (as a surrogate for climate change) on the hydrology of a fen peatland, Canada, Hydrological Processes, 20, 3589– 3600, 2006.

Wilson, P. G.: The relationship among micro-topographic variation, water table depth and biogeochemistry in an ombrotrophic bog, Master Thesis, Department of Geography McGill

University, Montreal, Quebec, p. 103, 2012.
Wojtuń B., Sendyk A. and Martynia, D.: Sphagnum species along environmental gradients in
mires of the Sudety Mountains (SW Poland), Boreal Environment Research, 18, 74–88, 2003.
Wu, J. and Roulet, N. T.: Climate change reduces the capacity of northern peatlands to absorb
the atmospheric carbon dioxide: The different responses of bogs and fens. Global
Biogeochemical Cycles, doi.org/10.1002/2014GB004845, 2014.
Wullschleger, S. D., Epstein, H. E., Box, E. O., Euskirchen, E. S., Goswami, S., Iversen, C. M.,
Kattge, J., Norby, R. J., van Bodegom, P. M. and Xu, X.: Plant functional types in Earth system
models: past experiences and future directions for application of dynamic vegetation models in
high-latitude ecosystems, Ann. Bot., 114, 1–16, 2014.

Table. 1 List of symbols and abbreviations

| Symbol | Description | Unit |
|---|---|---|
| $A$ | Net photosynthesis rate | µmol m$^{-2}$ s$^{-1}$ |
| $A_m$ | Maximal net photosynthesis rate | µmol m$^{-2}$ s$^{-1}$ |
| $\alpha_{imm}$ | Temperature constant for NSC immobilization | |
| $\alpha_{PPFD}$ | Half-saturation point of PPFD for photosynthesis. | µmol m$^{-2}$ s$^{-1}$ |
| $B_{cap}$ | Capitulum biomass | g m$^{-2}$ |
| $C_T$ | Specific heat | J K$^{-1}$ kg$^{-1}$ |
| $D_S$ | Capitulum density | shoots cm$^{-2}$ |
| $dH$ | Annual height growth of *Sphagnum* mosses | cm |
| $dWT$ | Hummock-lawn differences in water table | cm |
| $E$ | Rate of evaporation | cm timestep$^{-1}$ |
| $f_W$ | Water content multiplier on photosynthesis rate | |
| $f_T$ | Temperature multiplier on photosynthesis rate | |
| $h$ | Water potential | cm |
| $Hc$ | Shoot height of *Sphagnum* mosses | cm |
| $H_{cap}$ | Height of capitula | cm |
| $H_{spc}$ | Biomass density of living *Sphagnum* stems | g m$^{-2}$ cm$^{-1}$ |
| $I$ | Rate of net inflow water | cm |
| $k_{imm}$ | Specific immobilization rate | g g$^{-1}$ |
| $JD_{thaw}$ | Julian day when thawing completed | |
| $K_h$ | Hydraulic conductivity of peat layer | cm s$^{-1}$ |
| $K_m$ | Hydraulic conductivity of moss layer | cm s$^{-1}$ |

| | | |
|---|---|---|
| $K_{sat}$ | Saturated hydraulic conductivity | cm s$^{-1}$ |
| $K_T$ | Thermal conductivity | W m$^{-1}$ K$^{-1}$ |
| $lc$ | Width of a grid cell in simulation | cm |
| $M_B$ | Immobilized NSC to biomass production | g |
| $NSC_{max}$ | Maximal NSC concentration in *Sphagnum* biomass | g g$^{-1}$ |
| $P$ | Precipitation | cm |
| $Pm$ | Mass-based rate of maximal gross photosynthesis | μmol g$^{-1}$ s$^{-1}$ |
| $PPFD$ | Photosynthetic photon flux density | μmol m$^{-2}$ s$^{-1}$ |
| $\rho_{bulk}$ | Bulk density of peat | g cm$^{-3}$ |
| $r_{aero}$ | Aerodynamic resistance | s m$^{-1}$ |
| $r_{bulk}$ | Cell-level bulk surface resistance | s m$^{-1}$ |
| $r_{ss}$ | Bulk surface resistance of community | s m$^{-1}$ |
| $Rh$ | Relative humidity | % |
| $Rs$ | Mass-based respiration rate | μmol g$^{-1}$ s$^{-1}$ |
| $R_s$ | Incoming shortwave radiation | W m$^{-2}$ |
| $R_l$ | Incoming longwave radiation | W m$^{-2}$ |
| $S_c$ | Area of a cell in model simulation | m$^2$ |
| $s_{imm}$ | Multiplier for temperature threshold | |
| $Sv_i$ | Evaporative area of a cell $i$ | cm$^2$ |
| $T$ | Capitulum temperature | ℃ |
| $Ta$ | Air temperature | ℃ |
| $T_{opt}$ | reference temperature of respiration (20 ℃) | ℃ |
| $u$ | Wind speed | m s$^{-1}$ |

| | | |
|---|---|---|
| $W_{cap}$ | Capitulum water content | g g$^{-1}$ |
| $W_{cmp}$ | Capitulum water content at the compensation point | g g$^{-1}$ |
| $W_{max}$ | Maximum water content of capitula | g g$^{-1}$ |
| $W_{opt}$ | Optimal capitulum water content for photosynthesis | g g$^{-1}$ |
| $W_{cf}$ | field-water contents of *Sphagnum* capitulum | g g$^{-1}$ |
| $W_{sf}$ | field-water contents of *Sphagnum* stem | g g$^{-1}$ |
| $WTm$ | Measured multi-year mean of weekly water table | cm |
| $WTs$ | Simulated multi-year mean of weekly water table | cm |
| $z_m$ | Thickness of the living moss layer | cm |
| $\theta_m$ | Volumetric water content of moss layer | |
| $\theta_r$ | permanent wilting point water content | |
| $\theta_s$ | saturated water content | |

Abbreviations:

| | |
|---|---|
| $\Gamma$ | Learning rate of gradient decedent algorithms |
| D-layer | Daily-based snow layer |
| ICOS | Integrated Carbon Observation System |
| JD | Julian day |
| NSC | Nonstructural carbon |
| PMS | Peatland Moss Simulator |
| RWC | Capitulum water content |
| SD | Standard deviation |
| SE | Standard error |
| SSE | Sum of squared error |

| SVAT | Soil-vegetation-atmosphere transport |
| WT | Water table |


Table. 2 Species-specific values of morphological and photosynthetic parameters for *S.*
*magellanicum* and *S. fallax*. The parameters include: capitulum density ($D_S$), capitulum biomass
($B_{cap}$), specific height of stem ($H_{spc}$), maximal gross photosynthesis rate at 20 ℃ ($Pm_{20}$),
respiration rate at 20 ℃ ($Rs_{20}$), half-saturation point of photosynthesis ($α_{PPFD}$), and polynomial
coefficients ($a_{W0}$, $a_{W1}$ and $a_{W2}$) for the responses of net photosynthesis to capitulum water
content. Parameter values are given as mean ± standard deviation.

| Parameter | Unit | *S. magellanicum* | *S. fallax* | Equation |
|---|---|---|---|---|
| $D_S$ | $cm^{-2}$ | 0.922±0.289 | 1.46±0.323 | -[a] |
| $B_{cap}$ | $g\ m^{-2}$ | 75.4±21.5 | 69.2±19.6 | -[a] |
| $H_{spc}$ | $g^{-1}\ cm^{-1}$ | 45.4 ± 7.64 | 32.6±6.97 | (7) |
| $Pm_{20}$ | $μmol\ g^{-1}\ s^{-1}$ | 0.0189±0.00420 | 0.0140±0.00212 | (2) |
| $Rs_{20}$ | $μmol\ g^{-1}\ s^{-1}$ | 0.00729±0.00352 | 0.00651±0.00236 | (2) |
| $α_{PPFD}$ | $μmol\ m^{-2}\ s^{-1}$ | 101.4±14.1 | 143±51.2 | (2) |
| $a_{W0}$ | unitless | -1.354±0.623 | -1.046±0.129 | (4) |
| $a_{W1}$ | unitless | 0.431±0.197 | 0.755±0.128 | (4) |
| $a_{W2}$ | unitless | -0.0194±0.0119 | -0.0751±0.0223 | (4) |

[a] the parameter was used in the linear models predicting the $\log_{10}$-transformed capitulum water
potential (*h*) and bulk resistance ($r_{bulk}$) for *S. fallax* and *S. magellanicum*. The capitulum density
and photosynthetic parameter values measured here are well within the range of those reported in
literature for these species (McCarter & Price, 2014; Laing et al. 2014; Bengtsson et al. 2016;
Korrensalo et al. 2016).
Table 3. Parameters values for SVAT simulations (Module III). The parameters include:
saturated hydraulic conductivity ($K_{sat}$), water retention parameters of water retention curves ($\alpha$
and $n$), saturated water content ($\theta_s$), permanent wilting point water content ($\theta_r$), snow layer
surface albedos ($a_s$, $a_l$), the thermal conductivity ($K_T$), specific heat ($C_T$), maximal nonstructural
carbon (NSC) concentration ($NSC_{max}$).

| Parameter | Value | Equation | Source |
|---|---|---|---|
| $K_{sat}$ | 162 | A6 | McCarter and Price, 2014 |
| $n$ | 1.43 | A5 | McCarter and Price, 2014 |
| $\alpha$ | 2.66 | A5 | McCarter and Price, 2014 |
| $\theta_s$ | 0.95[a] | A5 | Päivänen, 1973 |
| $\theta_r$ | 0.071[b] | A5 | Weiss et al., 1998 |
| $a_s$ | 0.15 | A9 | Runkle et al., 2014 |
| $a_l$ | 0.02 | A10 | Thompson et al., 2015 |
| $K_{T,water}$ | 0.57 | A4 | Letts et al., 2000 |
| $K_{T,ice}$ | 2.20 | A4 | Letts et al., 2000 |
| $K_{T,org}$ | 0.25 | A4 | Letts et al., 2000 |
| $C_{T,water}$ | 4.18 | A3 | Letts et al., 2000 |
| $C_{T,ice}$ | 2.10 | A3 | Letts et al., 2000 |
| $C_{T,org}$ | 1.92 | A3 | Letts et al., 2000 |
| $NSC_{max}$ | 0.045 | 6 | Turetsky et al., 2008 |

[a] The value was calculated from bulk density ($\rho_{bulk}$) as $\theta_s = 97.95 - 79.72\rho_{bulk}$ following Päivänen
(1973); [b] The value was calculated as $\theta_r = 4.3 + 67\rho_{bulk}$ following Weiss et al. (1998).
Table 4. Results from the Test 2 addressing the robustness of the model to the uncertainties in a
set of parameters. Each parameter was increased or decreased by 10%. Model was run for *S.
magellanicum* and *S. fallax* in their preferential habitats. Difference in mean cover between
simulations under changed and unchanged parameter values are given with the standard
deviations (SD) of the means in brackets. The parameters include: specific immobilization rate
(*kimm*), maximal nonstructural carbon (NSC) concentration (*$NSC_{max}$*), hydraulic conductivity of
moss layer (*$K_m$*), hydraulic conductivity of peat layer (*$K_h$*), water retention parameters of water
retention curves (α and *n*), snow layer surface albedo (*$a_s$*) and aerodynamic resistance (*$r_{aero}$*).

| Change in parameter value | Equation | Changes in simulated cover, % (SD) | |
| --- | --- | --- | --- |
| | | *S. magellanicum* (hummock) | *S. fallax* (lawn) |
| *kimm* +10% | 5 | -1.2 (3.5) | -3.5 (3.8) |
| *kimm* -10% | | +2.7 (0.4) | -5.0 (3.4) |
| *$NSC_{max}$* +10% | 6 | +4.5 (2.9) | +0.7 (3.0) |
| *$NSC_{max}$* -10% | | -0.7 (4.0) | -4.8 (4.5) |
| *$K_m$* +10% | 10 | +1.0 (3.1) | -1.7 (2.3) |
| *$K_m$* -10% | | -1.7 (2.7) | +4.1 (4.3) |
| *$K_h$* +10% | A1 | -1.1 (3.0) | +1.1 (2.0) |
| *$K_h$* -10% | | -1.8 (3.1) | -0.5 (2.7) |
| *n* +10% | A5 | -1.6 (3.2) | -3.2 (3.2) |
| *n* -10% | | -9.4 (3.6) | -0.3 (2.9) |
| α +10 % | A5 | -0.5 (2.9) | -0.3 (2.3) |
| α -10 % | | -1.3 (3.6) | +3.2 (1.0) |
| *$a_s$* +10% | A9 | -2.2 (3.8) | +0.6 (2.1) |
| *$a_s$* -10% | | +3.3 (3.4) | +1.2 (1.8) |
| *$r_{aero}$* +10% | A14, A15 | -2.1 (3.4) | +0.3 (2.1) |
| *$r_{aero}$* -10% | | -3.8 (4.4) | +2.3 (1.1) |

Table 5. Result from the Test 7-10 addressing the importance of meteorological fluctuations,
stochasticity of model parameters and the photosynthetic water-response. In Test 7, monthly
mean values of meteorological variables were used to drive the model simulation. In Test 8, the
stochasticity of model parameters was removed, and average values were used to parameters at
grid cell level. In Test 9-10, the photosynthetic water-response parameters (i.e. $a_{W0}$, $a_{W1}$ and $a_{W2}$.
See Table 2) were set to be the same as those in *S. magellanicum* (Test 9) and same as those in *S.
fallax* (Test 10). The mean cover of *S. magellanicum* on hummocks and *S. fallax* on lawns after
the simulation of 15 year periods are listed in the table.

| Test | *S. magellanicum* (hummock) | *S. fallax* (lawn) |
|------|------------------------------|---------------------|
| 7    | 73%                          | 96%                 |
| 8    | 90%                          | 72%                 |
| 9    | 14 %                         | 100 %               |
| 10   | 100 %                        | 100 %               |


## Appendix A. Calculating community SVAT scheme (Module III)

*Transport of water and heat in peat profile*

Simulating the transport of water and heat in the peat profiles was based on Gong et al. (2012, 2013). Here we list the key algorithms and parameters. Ordinary differential equations governing the vertical transport of water and heat in peat profiles were given as:

$$C_h \frac{\partial h}{\partial t} = \frac{\partial}{\partial z}\left[K_h\left(\frac{\partial h}{\partial z} + 1\right)\right] + S_h \qquad (A1)$$

$$C_T \frac{\partial T}{\partial t} = \frac{\partial}{\partial z}\left(K_T \frac{\partial T}{\partial z}\right) + S_T \qquad (A2)$$

where $t$ is the time step; $z$ is the thickness of peat layer; $h$ is the water potential; $T$ is the temperature; $C_h$ and $C_T$ are the specific capacity of water (i.e. $\partial\theta/\partial h$) and heat; $K_h$ and $K_T$ are the hydraulic conductivity and thermal conductivity, respectively; and $S_h$ and $S_T$ are the sink terms for water and energy, respectively.

$C_T$ and $K_T$ were calculated as the volume-weighted sums from components of water, ice and organic matter:

$$C_T = C_{water}\theta_{water} + C_{ice}\theta_{ice} + C_{org}(1 - \theta_{water} - \theta_{ice}) \qquad (A3)$$

$$K_T = K_{water}\theta_{water} + K_{ice}\theta_{ice} + K_{org}(1 - \theta_{water} - \theta_{ice}) \qquad (A4)$$

where $C_{water}$, $C_{ice}$ and $C_{org}$ are the specific heats of water, ice and organic matter, respectively; $K_{water}$, $K_{ice}$ and $K_{org}$ are the thermal conductivities of water, ice and organic matter, respectively; and $\theta_{water}$ and $\theta_{ice}$ are the volumetric contents of water and ice, respectively.

For a given $h$, $C_h=\partial\theta(h)/\partial h$ was derived from the van Genuchten water retention model (van Genuchten, 1980) as:

$$\theta(h) = \theta_r + \frac{(\theta_s - \theta_r)}{[1 + (\alpha|h^n|)^m]} \qquad (A5)$$

where $\theta_s$ is the saturated water content; $\theta_r$ is the permanent wilting point water content; $\alpha$ is a scale parameter inversely proportional to mean pore diameter; $n$ is a shape parameter; and $m=1-1/n$.

Hydraulic conductivity ($K_h$) in an unsaturated peat layer was calculated as a function of $\theta$ by combining the van Genuchten model with the Mualem model (Mualem, 1976):

$$K_h(\theta) = K_{sat}S_e^{Le}\left[1 - \left(1 - S_e^{1/m}\right)^m\right] \qquad (A6)$$

where $K_{sat}$ is the saturated hydraulic conductivity; $S_e$ is the saturation ratio and $S_e = (\theta-\theta_r)/(\theta_s-\theta_r)$;
and $L_e$ is the shape parameter ($L_e$=0.5; Mualem, 1976).

*Boundary conditions and surface energy balance*
A zero-flow condition was assumed at the lower boundary of the peat column. The upper
boundary condition was defined by the surface energy balance, which was driven by net
radiation (*Rn*). The dynamics of *Rn* at surface x (*x*=0 for vascular canopy and *x*=1 for moss
surface) was determined by the balance between incoming and outgoing radiation components:
$Rn_x = Rsn_{b,x} + Rsn_{d,x} + Rln_x$         (A7)
where $Rsn_{b,x}$ and $Rsn_{d,x}$ are the absorbed energy from direct and diffuse radiation; $Rln_x$ is the
absorbed net longwave radiation.
Algorithms for calculating the net radiation components were detailed in Gong et al. (2013), as
modified from the methods of Chen et al. (1999). Canopy light interception was determined by
the light-extinction coefficient ($k_{light}$), leaf area index (*Lc*) and solar zenith angle. The
partitioning of reflected and absorbed irradiances at ground surface was regulated by the surface
albedos for the shortwave ($a_s$) and longwave ($a_l$) components, and the temperature of surface *x*
($T_x$) also affects net longwave radiation:
$Rn_x = Rsn_{b,x} + Rsn_{d,x} + Rln_x$         (A8)
$Rsn_{d,x} = Rs_{id,x}(1 - a_s)$         (A9)
$Rln_x = Rl_{i,x}(1 - a_l) - \varepsilon\delta T_x^4$

1003        (A10)

where $Rs_{ib}$, $Rs_{id}$, $Rl_i$ are the incoming beam, diffusive and longwave radiations; $\varepsilon$ is the emissivity
($\varepsilon = 1-a_l$); $\delta$ is the Stefan Boltzmann's constant ($5.67\times10^{-8}$ W m$^{-2}$ K$^{-4}$).
$Rn_x$ was partitioned into latent heat flux ($\lambda E_x$), sensible heat flux ($H_x$) and ground heat flux (for
canopy $G_1$=0):
$Rn_x = H_x + \lambda E_x + G_x$

1009        (A11)

$G_1 = K_T (T_x - Ts)/(0.5z)$         (A12)
where *Ts* is the temperature of the moss carpet; *z* is the thickness of the moss layer ($z = 5$ cm).
The latent heat flux was calculated by the "interactive scheme" (Daamen and McNaughton,
2000; see also in Gong et al., 2016), which is a K-theory-based, multi-source model:
$\quad \lambda E_x = \frac{(\Delta/\gamma)A_x r_{sa,x} + \lambda VPD_b}{r_{b,x} + (\Delta/\gamma)r_{sa,x}}$ (A13)
where $\Delta$ is the slope of the saturated vapor pressure curve against air temperature; $\lambda$ is the latent
heat of vaporization; $E$ is the evaporation rate; $VPD_b$ is the vapor pressure deficit at $z_b$; $r_{b,x}$ is the
total resistance to water vapor flow, the sum of boundary layer resistance ($r_{sa,x}$) and surface
resistance ($r_{ss}$); and $A$ is the available energy for evapotranspiration and $A_x = Rn_x - G_x$.
The calculations of $\gamma$, $\lambda$ and $VPD_b$ require the temperature ($T_b$) and vapor pressure ($e_b$) at the
mean source height ($z_b$). These variables were related to the total of latent heat ($\sum \lambda E_x$) and
sensible heat ($\sum H_x$) from all surfaces using the Penman-type equations:
$\quad \Sigma \lambda E_x = \rho_a C_p \, (e_b - e_a)/(r_{aero}\gamma)$ (A14)
$\quad \Sigma H_x = \rho_a C_p \, (T_b - T_a)/r_{aero}$
$\qquad$ (A15)
where $\rho_a C_p$ is the volumetric specific heat of air; $r_{aero}$ is the aerodynamic resistance between $z_b$
and the reference height $z_a$, and was a function of $T_b$ accounting for the atmospheric stability
(Choudhury and Monteith, 1988); and $\gamma$ is the psychrometric constant ($\gamma = \rho_a Cp/\lambda$).
Changes in the energy balance affect the surface temperature ($T_x$) and vapor pressure ($e_x$), which
further feed back to the energy availability (Eq. A10, A12), the source-height temperature, $VPD$
and the resistance parameters (e.g., $r_{aero}$). The values of $T_x$ and $e_x$ were solved iteratively by
coupling the energy balance equations (eqs. A11–A15) with the Penman-type equations (see also
Appendix B in Gong et al., 2016):
$\quad \lambda E_x = \rho_a C_p \, (e_x - e_b)/(r_{sa,x}\gamma)$ (A16)
$\quad H_x = \rho_a C_p \, (T_x - T_b)/r_{sa,x}$ (A17)
where the boundary-layer resistance for ground surface ($r_{sa,1}$) and canopy ($r_{sa,0}$) were calculated
following the approaches of Choudhury and Monteith (1988).

*Sink terms of transport functions for water and heat*
The sink term $S_{h,i}$ (see Eq. A11) for each soil layer $i$ was calculated as:
$\quad S_{h,i} = E_i - P_i - W_{melt,i} - I_i$ (A18)
where $E_i$ is the evaporation loss of water from the layer; $P_i$ is rainfall ($P_i = 0$ if the layer is not
topmost, i.e. $i>1$); $W_{melt,i}$ is the amount of melt water added to the layer; $I_i$ is the net water inflow
and was calibrated in Section 2.5.
The value of $E_i$ was calculated as:
$E_i = f_{top}E_0 + f_{root}(i)E_1$ (A19)
where $E_0$ and $E_1$ are the evaporation rate from ground surface and canopy (Eq. A13); $f_{top}$ is the
location multiplier for the topmost layer ($f_{top} = 0$ in cases $i>1$); and $f_{root}(i)$ is the fraction of fine-
root biomass in layer $i$.
The value of $W_{melt,i}$ was controlled by the freeze-thaw dynamics of soil water and snow pack,
which were related to the heat diffusion in soil profile (Eq. A2). We set the freezing point
temperature to 0 ℃, and the temperature of a soil layer was held constant (0 ℃) during freezing
or melting. For the $i$th soil layer, the sink term ($S_T$) in heat transport equation (Eq. A2) was
calculated as:
$S_{T,i} = f_{phase}max(|T_i|C_{T,i}, W_{phase}\lambda_{melt})$ (A20)
where $C_{T,i}$ is specific heat of soil layer (Eq. A13); $W_{phase}$ is the water content for freezing ($W_{phase}$
$= \theta_w$) or melting ($W_{phase} = \theta_{ice}$); $\lambda_{melt}$ is the latent heat of freezing; $f_{phase}$ is binarial coefficient that
denotes the existence of freezing or thawing. For each time step $t$, we computed $T_i(t)$ with a piror
assumption that $S_{T,i}=0$. Then $f_{phase}$ was determined by whether the temperature changed across
the freezing point, i.e. $f_{phase}=1$ if $T_i(t)*T_i(t-1) \leq 0$, otherwise $f_{phase}= 0$.

*Parameterization of SVAT processes*
For the calculation of surface energy balance, we set the height and leaf area of vascular
canopy to 0.4 m and 0.1 m$^2$ m$^{-2}$, consistent with the scarcity of vascular canopies at the site. The
aerodynamic resistance ($r_{aero}$, Eq. A14, Appendix A) for surface energy fluxes was calculated
following Gong et al. (2013a). The bulk surface resistance of community ($r_{ss}$, Eq. A13, Appendix
A) was summarized from the cell-level values of $r_{bulk,i}$, that $1/r_{ss} = \sum(1/r_{bulk,i})$. To calculate the
peat hydrology and water table, peat profiles of hummock and lawn communities were set to 150
cm deep and stratified into horizontal layers of depths varying from 5cm (topmost) to 30cm
(deepest). For each peat layer, the thermal conductivity ($K_T$) of fractional components, i.e. peat,
water and ice, were evaluated following Gong et al. (2013a). The bulk density of peat ($\rho_{bulk}$) was
set to 0.06 g cm$^{-3}$ below acrotelm (40 cm depth, Laine et al., 2004), and decreased linearly
toward the living moss layer. The saturated hydraulic conductivity ($K_{sat}$, Eq. A6, Appendix A)
and water retention parameters (i.e. α and $n$, Eq. A5, Appendix A) of water retention curves were
calculated as functions of $\rho_{bulk}$ and the depth of peat layer following Päivänen (1973). $K_{sat}$, $\alpha$ and
$n$ for the living moss layer were adopted from the values measured by McCarter and Price (2014)
from *S. magellanicum* carpet. The parameter values for SVAT processes are listed in Table 3.
*Calculation of snow dynamics*
In boreal and arctic regions, the amount and timing of snow melt has crucial impact on moisture
conditions, especially at fen peatlands. Therefore, to have realistic spring conditions we
introduced a snow-pack model, SURFEX v7.2 (Vionnet et al., 2007), into the SVAT modelling.
The snow-pack model simulates snow accumulation, wind drifting, compaction and changes in
metamorphism and density. These processes influenced the heat transport and freezing-melting
processes (i.e. $S_h$ and $S_T$, see Eq. A1-A2, Appendix A). In this modelling, we calculate the snow
dynamics on a daily basis in parallel to the SVAT simulation. Daily snowfall was converted into
a snow layer and added to ground surface. For each of the day-based snow layers (D-layers), we
calculated the changes in snow density, particle morphology and layer thicknesses. At each time
step, D-layers were binned into layers of 5-10 cm depths (S-layers) and placed on top of the peat
column for SVAT modelling. With a snow layer present, surface albedos (i.e. $a_s$, $a_l$) were
modified to match those of the topmost snow layer (see Table 4 in Vionnet et al., 2007). If the
total thickness of snow was less than 5 cm, all D-layers were binned into one S-layer. The
thermal conductivity ($K_T$), specific heat ($C_T$), snow density, thickness and water content of each
S-layer were calculated as the mass-weighted means from the values of D-layers. Melting and
refreezing tended to increase the density and $K_T$ of a snow layer but decrease its thickness (see
Eq. 18 in Vionnet et al., 2007). The fraction of melted water that exceeded the water holding
capacity of a D-layer (see Eq. 19 in Vionnet et al., 2007) was removed immediately as
infiltration water. If the peat layer underneath was saturated, the infiltration water was removed
from the system as lateral discharge.
*Boundary conditions and driving variables*
A zero-flow boundary was set at the bottom of peat. At peat surface the boundary conditions of
water and energy were defined by the ground surface temperature ($T_0$, see Eq. A10-A15 in
Appendix A) and the net precipitation ($P$ minus $E$). The profiles of layer thicknesses, $\rho_{bulk}$ and
hydraulic parameters were assumed to be constant during simulation. Lateral boundary
conditions were used to calculate the spreading of *Sphagnum* shoots among cells along the edge
of the model domain so that shoots can spread across the edge of simulation area and invade into
the grid cell at the boarder of the opposite side.
The model simulation was driven by climatic variables of air temperature ($Ta$), precipitation
(P), relative humidity ($Rh$), wind speed ($u$), incoming shortwave radiation ($Rs$) and longwave
radiation ($Rl$). To support the stochastic parameterization of the model and Monte-Carlo
simulations, Weather Generator (Strandman et al., 1993) was used to generate randomized
scenarios based on long-term weather statistics (period of 1981-2010) from the four closest
weather stations of the Finnish Meteorological Institute. This generator had been intensively
tested and applied under Finnish conditions (Kellomäki and Väisänen, 1997; Venäläinen et al.,
2001; Alm et al., 2007). We also compared the simulated meteorological variables against 2-year
data measured from Siikaneva peatland site (61°50 N; 24°10 E), located 10 km away from our
study site (Appendix C).

 **Appendix B. Methods and results of the empirical study on *Sphagnum* capitula water**
 **retention as a controlling mechanism for peatland moss community dynamics**


*Measurement of morphological traits*
To quantify morphological traits, samples of *S. fallax* and *S. magellanicum* were collected at the
end of August 2016 with a core (size d 7cm, area 50 cm$^2$, height at least 8 cm) maintaining the
natural density of the stand. Samples were stored in plastic bags at cool room (4 ℃) until
measurements. Eight replicates were collected for each species. For each sample, capitulum
density ($D_S$, shoots cm$^{-2}$) was measured and ten moss shoots were randomly selected and
separated into capitula and stems (5 cm below capitula). The capitula and stems were moistened
and placed on top of a tissue paper for 2 minutes to extract free-moving water, before weighing
them for water-filled fresh weight. The samples were dried at 60 °C for at least 48h to measure
the dry masses. The field-water contents of capitula ($W_{cf}$, g g$^{-1}$) and stems ($W_{sf}$, g g$^{-1}$) were then
calculated as the ratio of water to dry mass for each sample. The biomass of capitula ($B_{cap}$, g m$^{-2}$
$^{2}$) and living stems ($B_{st}$, g m$^{-2}$) were calculated by multiplying the dry masses with the capitulum
density ($D_S$). Biomass density of living stems ($H_{spc}$, g cm$^{-1}$ m$^{-2}$) was calculated by dividing $B_{st}$
with the length of stems.
*Measurement of photosynthetic traits*
We measured the photosynthetic light response curves for *S. fallax* and *S. magellanicum* with
fully controlled, flow-through gas-exchange fluorescence measurement systems (GFS-3000,
Walz, Germany; Li-6400, Li-Cor, US) under varying light levels. In 2016, measurements on
field-collected samples were done during May and early June, which is a peak growth period for
*Sphagna* (Korrensalo et al. 2017). Samples were collected from the field site each morning and
were measured the same day at Hyytiälä field station. Samples were stored in plastic containers
and moistened with peatland water to avoid changes in plant status during the measurement.
Right before the measurement we separated *Sphagnum* capitula from their stems and dried them
lightly using tissue paper before placing an even layer of them in a custom-made cuvette by
retaining the same density as naturally at field (Korrensalo et al. 2017). Net photosynthesis rate
($A$, µmol g$^{-1}$ s$^{-1}$) was measured at 1500, 250, 35, and 0 µmol m$^{-2}$ s$^{-1}$ photosynthetic photon flux
density ($PFD$) (Fig 1B). The light levels were chosen based on previous investigation by Laine
et al. (2011, 2015), which showed increasing A until PPFD at 1500 and no photoinhibition even
at high values of 2000 µmol m$^{-2}$ s$^{-1}$. The samples were allowed to adjust to cuvette conditions
before the first measurement and after each change in the PPFD level until the $CO_2$ rate had
reached a steady level, otherwise the cuvette conditions were kept constant (temperature 20°C,
$CO_2$ concentration 400 ppm, flow rate 500 umol s$^{-1}$, impeller at level 5 and relative humidity of
inflow air 60%, yet the relative humidity remained on average 81% during the measurements).
The time required for a full measurement cycle varied between 60 and 120 minutes. Each sample
was weighed before and after the gas-exchange measurement, then dried at 40°C for 48 h to
determine the biomass of capitula ($B_{cap}$). For each species, five samples were measured as
replicates and were made to fit a hyperbolic light-saturation curve (Larcher, 2003):
$$A_{20} = \left( \frac{Pm_{20}*PPFD}{\alpha_{PPFD}+PPFD} - Rs_{20} \right) * B_{cap} \tag{B1}$$
where subscript 20 denotes the variable value measured at 20 °C; $Rs$ is the mass-based dark
respiration rate ($\mu mol\ g^{-1}\ s^{-1}$); $Pm$ is the mass-based rate of maximal gross photosynthesis ($\mu mol$
$g^{-1}\ s^{-1}$); and $\alpha_{PPFD}$ is the half-saturation point ($\mu mol\ m^{-2}\ s^{-1}$), i.e., PPFD level where half of $Pm$ is
reached. The measured morphological and photosynthetic traits are listed in Table 2.

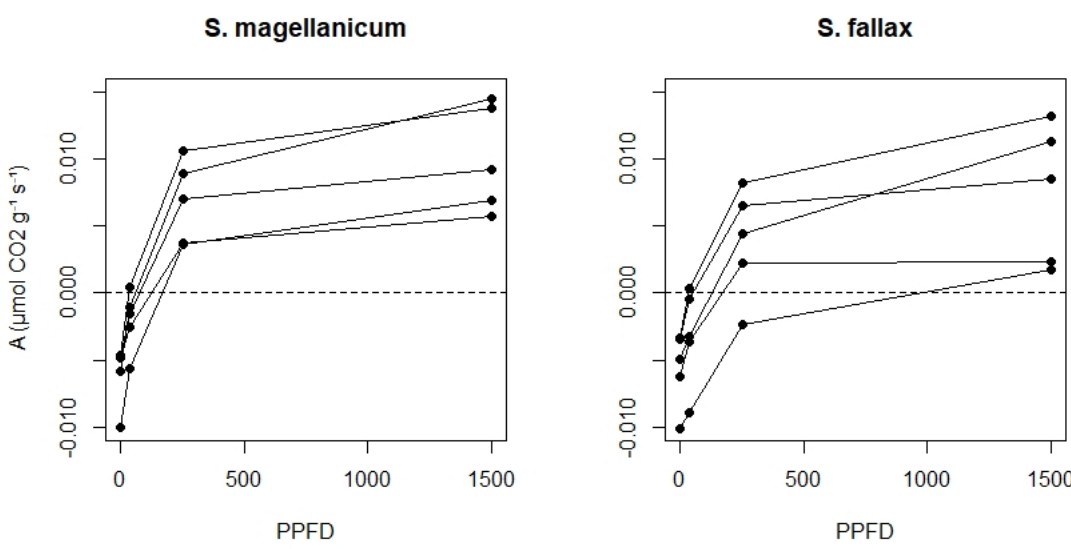


Figure B1. Measured light response curves for *S. magellanicum* and *S. fallax*.

*Drying experiment*
To link the water retention and photosynthesis of *Sphagnum* capitula, we performed a drying
experiment using a GFS-3000 system to measure co-variations of capitulum water potential (*h*,
cm water), water content ($W_{cap}$, $g\ g^{-1}$) and A ($\mu mol\ g^{-1}\ s^{-1}$). For both species, four mesocosms
were collected in August 2018 and transported to laboratory in UEF Joensuu, Finland. Capitula
were harvested and wetted by water from the mesocosms. The capitula were then placed gently
on a piece of tissue paper for 2 minutes and then placed into the same cuvette as used in the
previous photosynthesis measurement. The cuvette was then placed into GFS and measured
under constant conditions of *PPFD* (1500 umol m-2 s-1), temperature (293.2K), inflow air (700
umol s-1), $CO_2$ concentration (400 ppm) and relative humidity (40%). Measurement was stopped
when *A* dropped to less than 10% of its maximum. Each measurement lasted between 120
and180 minutes. Each sample was weighed before and after the gas-exchange measurement, then
dried at 40°C for 48 h to determine the biomass of capitula ($B_{cap}$).
The GFS-3000 records the vapor pressure ($e_a$, kPa) and the evaporation rate ($E$, g s$^{-1}$)
simultaneously with $A$ at every second (Heinz Walz GmbH, 2012). The changes in $W_{cap}$ with
time ($t$) was calculated as following:
$$RWC(t) = \left(W_{pre} - B_c - \sum_{t=0}^{t} E(t)\right)/B_c \tag{B2}$$
We assumed that the vapor pressure at the surface of water-filled cells equaled the saturation
vapor pressure ($e_s$), and the vapor pressure in the headspace of cuvette equaled that in the
outflow ($e_a$). The vapor pressure in capitula pores ($e_i$) thus can be calculated based on following
gradient-transport function (Fig. B2A):
$$\lambda E(t) = \frac{\rho_a C_p}{\gamma} \frac{(e_i(t) - e_a(t))}{r_a(t)} = \frac{\rho_a C_p}{\gamma} \frac{(e_s - e_i(t))}{r_s(t)} \tag{B3}$$
where $\lambda$ is the latent heat of vaporization; $\gamma$ is the slope of the saturation vapor pressure -
temperature relationship; $r_a$ is the aerodynamic resistance (m s$^{-1}$) for vapor transport from inter-
leaf volume to headspace; $r_s$ is the surface resistance of vapor transport from wet leaf surface to
inter-leaf volume. The bulk resistance for evaporation ($r_{bulk}$) was thus calculated as $r_a + r_s$.
We assumed that the structures of tissues and pores did not change during the drying process
and assumed $r_a$ to be constant during each measurement. $A$ tended to increase with time $t$ until it
peaked ($A_m$) and then decreased (Fig. 2B). The point $A=A_m$ implied the water content where
further evaporative loss would start to drain the cytoplasmic water, leading to the decrease in $A$.
The response of $A$ to $W_{cap}$ was fitted as a second-order polynomial function (Robroek et al.,
2009) using data from $t_{Am}$ to $t_n$:
$f_A(W_{cap}) = a_{W0} + a_{W1} * W_{cap} + a_{W2} * W_{cap}{}^2$
1198          (B4)

where $a_{w0}$, $a_{w1}$ and $a_{w2}$ are parameters; and $f_A(W_{cap}) = A/A_m$. For each replicate, the optimal water
content for photosynthesis ($W_{opt}$) was derived from the peak of fitted curve (Eq. 4). The
capitulum water content at the compensation point $W_{cmp}$, where the rates of gross photosynthesis
and respiration are equal, can be calculated from the point $A=0$.





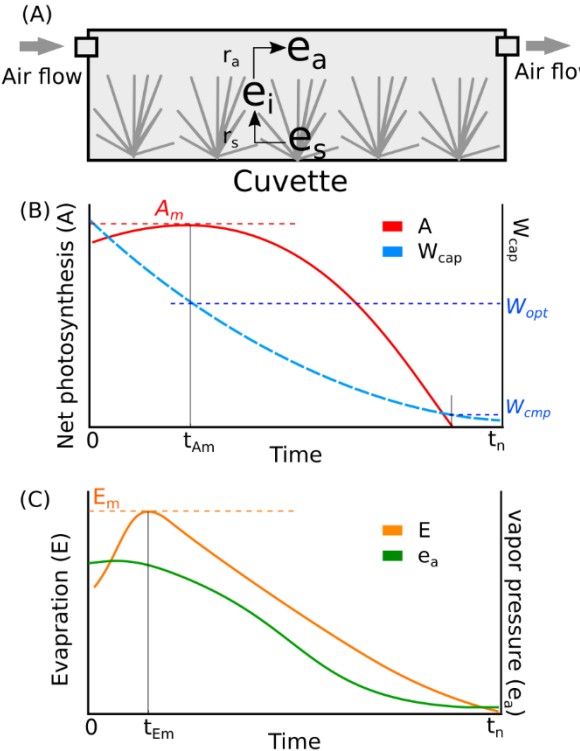

Figure B2. Conceptual schemes of (A) cuvette setting and resistances, (B) the co-variations of net photosynthesis and $W_{cap}$, and (C) the co-variations of evaporation and vapor pressure in headspace during a measurement. Meanings of symbols: $e_a$, vapor pressure in headspace of cuvette (kPa); $e_i$, vapor pressure in branch-leaf structure of capitula; $e_s$, vapor pressure at the surface of wet tissues; $r_a$, aerodynamic resistance of vapor diffusion from inner capitula to headspace; $r_s$, surface resistance of vapor diffusion from wet tissue surface to inner capitula space; $A$, net photosynthesis rate ($\mu$mol m$^{-2}$ s$^{-1}$); $A_m$, maximal net photosynthesis rate ($\mu$mol m$^{-2}$ s$^{-1}$); $W_{cap}$, water content of capitula (g g$^{-1}$); $W_{opt}$, $W_{cap}$ at $A=A_m$; $W_{cmp}$, $W_{cap}$ at $A=0$; $E$, evaporation rate (mm s$^{-1}$).



Similarly, the evaporation rate ($E$) increased from the start of measurement until maximum
evaporation $E_m$, and then decreased (Fig. B2C). The point $E=E_m$ implied the time when the wet
capitulum tissues were maximally exposed to the air flow. Therefore, $r_a$ was estimated as the
minimum of bulk resistance using Eq. (B5), by assuming $e_i(t) \approx e_s$ when $E(t) = E_m$:
$r_a = \frac{\rho_a C_p}{\gamma} \frac{(e_s - e_a(t))}{\lambda E_m}$                                    (B5)
Based on the calculated $e_i(t)$, we were able to derive the capitulum water potential ($h$)
following the equilibrium vapor-pressure method (e.g. Price et al, 2008; Goetz and Price, 2015):
$h = \frac{RT}{Mg} ln\left(\frac{e_i}{e_s}\right) + h_0$                                    (B6)
where $R$ is the universal gas constant (8.314 J mol$^{-1}$ K$^{-1}$); $M$ the molar mass of water (0.018 kg
mol$^{-1}$); $g$ is the gravitational acceleration (9.8 N kg$^{-1}$); $e_i/e_s$ is the relative humidity; $h_0$ is the
water potential due to the emptying of free-moving water before measurement (set to 10 kPa
according to Hayward and Clymo, 1982).

*Statistical analysis*
The light response curve (Eq. B1) and the response function of $A/A_m$ to $W_{cap}$ changes (Eq. B4)
were fitted using nlme package in R (version 3.1). The obtained values of shape parameters $a_{W0}$,
$a_{W1}$ and $a_{W2}$ (Eq. 4) were then used to calculate $W_{opt}$ ($W_{opt}$ = -0.5 $a_{W1}/ a_{W2}$) and $W_{cmp}$ ($W_{cmp}$ = 0.5
[-$a_{W1}$ - ($a_{W1}^2$ - $4a_{W0} a_{W2})^{0.5}$] / $a_{W2}$). We then applied ANOVA to compare *S. magellanicum* against
*S. fallax* for the traits obtained from the field sampling (i.e. structural properties such as $B_{cap}$, $D_S$,
$H_{spc}$, $W_{cf}$, $W_{sf}$) and from the gas-exchange measurements (i.e. $Pm_{20}$, $Rs_{20}$, $W_{opt}$, $W_{cmp}$ and $r_{bulk}$),
using R (version 3.1).
The measured values of capitulum water potential ($h$) were log$_{10}$-transformed and related to the
variations in $W_{cap}$, $B_{cap}$ and $D_S$ with a linear model. Similarly, a linear model was established to
quantify the response of bulk resistance for evaporation ($r_{bulk}$) (log$_{10}$-transformed) to the
variations in $h$, $B_{cap}$ and $D_S$. The linear regressions were based on statsmodels (version 0.9.0) in
Python (version 2.7), as supported by Numpy (version 1.12.0) and Pandas (version 0.23.4)
packages.

## 1255 Results of the empirical measurements

The two *Sphagnum* species differed in their structural properties (Table B1). Lawn species S.
*fallax* had looser structure than hummock species *S. magellanicum* as seen in lower capitulum
density ($D_S$) and specific height ($H_{spc}$) in *S. fallax* than in *S. magellanicum* (P<0.05, Table. B1).
Moreover, in conditions prevailing in the study site *S. fallax* mosses were dryer than *S.
magellanicum*; the field-water contents of *S. fallax* capitulum ($W_{cf}$) and stem ($W_{sf}$) were 40% and
46% lower than *S. magellanicum* (P<0.01, Table. B1), respectively. The different density of
capitulum of the two species differing in their capitulum size led to similar capitulum biomass
($B_{cap}$) (P=0.682) between *S. fallax* with small capitulum and *S. magellanicum* with large
capitulum. Unlike the structural properties, maximal $CO_2$ exchange rates ($Pm_{20}$ and $Rs_{20}$) did not
differ between the two species (Table B1).
The drying experiment demonstrated how capitulum water content regulated capitulum
processes in both studied *Sphagnum* species (Fig. B3). Decreasing capitulum water content
($W_{cap}$) led to decrease in the water potential ($h$), the responses of $h$ to $W_{cap}$ varied among
replicates (Fig. 3A). The values of $W_{cap}$ for *S. fallax* were generally lower than those for *S.
magellanicum* under the same water potentials. The fitted linear models explained over 95% of
the variations in the measured $h$ for both species (Table. B2), although fitted responses of $h$ to
$W_{cap}$ were slightly smoother than the measured ones, particularly for *S. magellanicum* (Fig.
B3A). The responses of $h$ to $W_{cap}$ was significantly affected by the capitulum density ($D_S$),
capitulum biomass ($B_{cap}$) and their interactions with $W_{cap}$ (Table. B2).
Decreasing capitulum water content ($W_{cap}$), and water potential ($h$), were associated with
increasing bulk resistance for evaporation ($r_{bulk}$, Fig. B3B), although the sensitivity of $r_{bulk}$ to $h$
changes varied by replicates. The values of $r_{bulk}$ from *S. fallax* were largely lower than those
from *S. magellanicum* when the capitulum water content of the two species were similar. The
fitted linear models explained the observed variations in the measured $r_{bulk}$ well for both species
(Fig. 2B and Table. B3). The variation in the response of $r_{bulk}$ to $h$ was significantly affected by
capitulum density ($D_S$), capitulum biomass ($B_{cap}$) and their interactions with $h$ (Table. B3).
Decreasing capitulum water content ($W_{cap}$) slowed down the net photosynthesis rate (Fig.
B2C), as represented by the decreasing ratio of A/A$_m$. *S. fallax* required lower capitulum water
content ($W_{cap}$) than *S. magellanicum* to reach photosynthetic maximum and photosynthetic
compensation point. However, the ranges of capitulum water content from photosynthetic
maximum ($W_{opt}$) or field capacity ($W_{fc}$) to that at compensation point ($W_{cmp}$) were smaller for *S.*
*fallax* than *S. magellanicum*. Hence, *S. fallax* had narrower transition zone for photosynthesis to
respond to drying, compared to *S. magellanicum*.

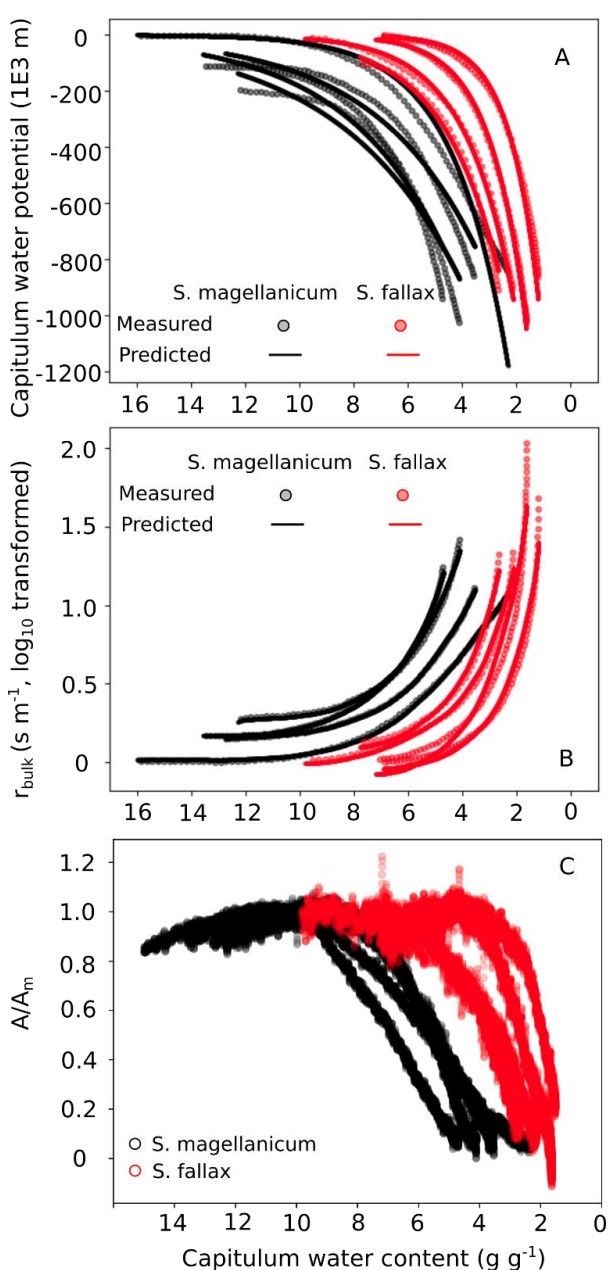


Figure B3. Responses of (A) capitulum water potential, (B) bulk resistance of evaporation, and (C) net photosynthesis to changes in capitulum water content ($W_{cap}$) of two *Sphagnum* species typical to hummocks (*S. magellanicum*, black) and lawns (*S. fallax*, red). As the measured results are based on the drying experiment starting with fully wetted capitula characteristic for both species, the X-axis is shown from high to low $W_{cap}$. The values predicted in (B) and (C) are based on linear models with parameter values listed in Tables B2 and B3 and predictor values from the drying experiment.

1299

Table. B1 Species-specific traits of morphological, photosynthetic and water-retention from *S. magellanicum* and *S. fallax*. Trait values (mean ± standard deviation) and ANOVA statistics F- and p-values are given for comparing the means of traits of the two species.

| Trait | *S. magellanicum* | *S. fallax* | F | P (>F) |
|---|---|---|---|---|
| Capitulum density, $D_S$ (capitula cm$^{-2}$) | 0.922±0.289 | 1.46±0.323 | 6.224[a] | 0.037 * |
| Capitulum biomass, $B_{cap}$ (g m$^{-2}$) | 75.4±21.5 | 69.2±19.6 | 0.181[a] | 0.682 |
| Specific height, $H_{spc}$ (cm g$^{-1}$ m$^{-2}$) | 45.4 ± 7.64 | 32.6±6.97 | 6.126[a] | 0.038* |
| Field water content of capitula, $W_{cf}$ (g g$^{-1}$) | 14.7±3.54 | 8.09±1.48 | 11.75[a] | 0.009** |
| Field water content of stems, $W_{sf}$ (g g$^{-1}$) | 18.4±1.92 | 10.2±1.50 | 45.81[a] | 0.0001** |
| Maximal gross photosynthesis rate at 20 ℃, $Pm_{20}$ (µmol g$^{-1}$ s$^{-1}$) | 0.019±0.004 | 0.014±0.002 | 3.737[b] | 0.101 |
| Respiration rate at 20 ℃, $Rs_{20}$ (µmol g$^{-1}$ s$^{-1}$) | 0.007±0.004 | 0.007±0.002 | 0.012[b] | 0.92 |
| half-saturation point of photosynthesis, $\alpha_{PPFD}$ (µmol m$^{-2}$ s$^{-1}$) | 101.4±14.1 | 143±51.2 | 2.856[b] | 0.142 |
| Optimal capitulum water content for photosynthesis, $W_{opt}$ (g g$^{-1}$) | 9.41±0.73 | 5.81±1.68 | 11.57[b] | 0.0145* |
| Capitulum water content at photosynthetic compensation point, $W_{cmp}$ (g g$^{-1}$) | 3.67±0.83 | 1.78±0.43 | 12.35[b] | 0.0126* |
| Minimal bulk resistance of evaporation, $r_a$ (m s$^{-1}$) | 33.5±7.30 | 40.7±4.99 | 1.976[b] | 0.2165 |

[a] soil-core measurement, sample $n=5$; [b] cuvette gas-exchange measurement, sample $n=4$; * the difference of means is significant (P<0.05); ** the difference of means is very significant (P<0.01).

Table B2. Parameter estimates of the linear model for the $\log_{10}$-transformed capitulum water
potential ($h$) for *S. fallax* and *S. magellanicum*. Estimate value, standard error (SE), and test
statistics p-values are given to the predictors of the models. Predictors are: capitulum biomass
($B_{cap}$), capitulum density ($D_S$), capitulum water content ($W_{cap}$), the interaction of capitulum
biomass and water potential ($B_{cap} \times W_{cap}$), the interactions of capitulum biomass and capitulum
density ($D_S \times W_{cap}$), the interactions of capitulum density and water potential ($D_S \times W_{cap}$), and the
interaction of capitulum biomass, capitulum density and water potential ($B_{cap} \times D_S \times W_{cap}$). All
coefficient values are significantly different from 0 ($p<0.001$).

| Parameter | *S. magellanicum* ($R^2$=0.972) | | *S. fallax* ($R^2$=0.984) | |
|---|---|---|---|---|
| | Value | SE | Value | SE |
| (Intercept) | 25.30 | 0.253 | -90.99 | 2.158 |
| $B_{cap}$ | -272.10 | 3.133 | 2294.67 | 52.342 |
| $W_{cap}$ | -9.50 | 0.031 | -62.12 | 0.600 |
| $B_{cap} \times W_{cap}$ | 114.61 | 0.387 | 1500.26 | 14.549 |
| $D_S$ | -21.76 | 0.253 | 104.11 | 2.376 |
| $B_{cap} \times D_S$ | 268.95 | 3.112 | -2422.79 | 55.251 |
| $D_S \times W_{cap}$ | 9.33 | 0.031 | 68.35 | 0.661 |
| $B_{cap} \times D_S \times W_{cap}$ | -113.33 | 0.386 | -1588.06 | 15.360 |




Table B3. Parameter estimates of the linear model for the $\log_{10}$-transformed capitulum
evaporative resistance ($r_{bulk}$) for *S. fallax* and *S. magellanicum*. Estimate value, standard error
(SE), and test statistics p-values are given to the predictors of the models. Predictors are:
capitulum biomass ($B_{cap}$), capitulum density ($D_S$), water potential ($h$), the interaction of
capitulum biomass and water potential ($B_{cap} \times h$), the interactions of capitulum biomass and
capitulum density ($D_S \times h$), the interactions of capitulum density and water potential ($D_S \times h$), and
the interaction of capitulum biomass, capitulum density and water potential ($B_{cap} \times D_S \times h$). All
coefficient values are significantly different from 0 ($p<0.001$).

| Parameter | *S. magellanicum* ($R^2=0.998$) | | *S. fallax* ($R^2=0.966$) | |
|---|---|---|---|---|
| | Value | SE | Value | SE |
| (Intercept) | -1.13 | 0.027 | 55.07 | 2.225 |
| $B_{cap}$ | 14.45 | 0.334 | 1334.55 | 53.968 |
| $h$ | 0.0012 | 5.92e-05 | -0.028 | 0.004 |
| $B_{cap} \times h$ | -0.0007 | 0.001 | 0.707 | 0.101 |
| $D_S$ | 1.08 | 0.027 | -60.53 | 2.450 |
| $B_{cap} \times D_S$ | -13.39 | 0.333 | 1406.36 | 56.968 |
| $D_S \times h$ | 0.0002 | 5.89e-05 | 0.0317 | 0.005 |
| $B_{cap} \times D_S \times h$ | -0.0017 | 0.001 | -0.733 | 0.106 |

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

 **Appendix C. Comparisons of meteorological variables simulated by Weather Generator**
**and those measured from Siikaneva peatland site (ICOS site located in 10 km distance**
**from the study site Lakkasuo)**

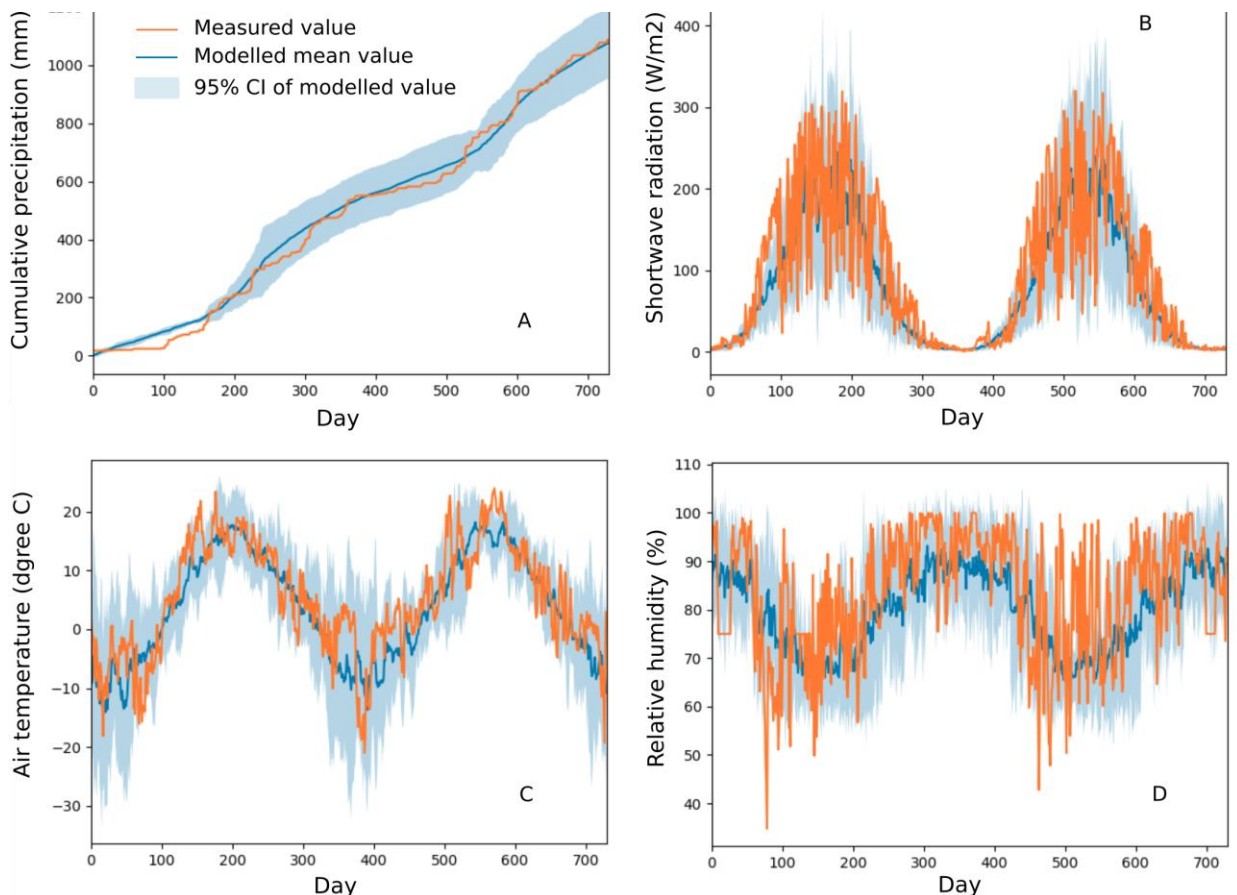

Fig. C1 Comparisons of meteorological variables simulated by Weather Generator and those measured from Siikaneva peatland site. The variables include (A) cumulative precipitation (mm), (B) incoming shortwave radiation (W m$^{-2}$), (C) air temperature (°C), and (D) relative humidity (%). These variables were measured and simulated at half-hourly timescale. The measurements were carried out during 2012-2013. Details about the site and measurements have been described by Alekseychik et al. (2018). The measured seasonal dynamics of the meteorological variables were generally in line with the 95% confidence intervals (CI) of the simulated values, which were calculated based on Monte-Carlo simulations (n=5).

**Appendix D. Comparisons of seasonal water table measured from the study site and the**
**values simulated based on calibrated net inflow**

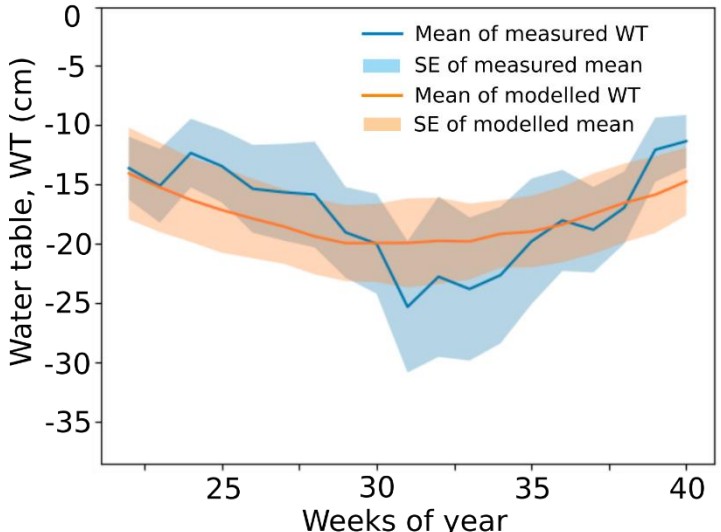


Fig. D1 Comparison of seasonal water table (WT) measured at the Lakkasuo study site and the
values simulated by the calibrated PCS. WT values were sampled weekly from the lawn habitats
both in field and in model output. The weekly mean WT was measured during 2001, 2002, 2004
and 2016. The modelled means and standard deviations (SD) of WT were based on 20 Monte-
Carlo simulations. The simulated seasonality of mean WT generally followed the measured
trends. The calibration reduced the sum of squared error ($SE$, Eq. 12) from 199.5 ($a_N=b_N=0$) to
117.3. The calibrated values for $a_N$ and $b_N$ were $-5.3575 \times 10^{-4}$ and $4.7599 \times 10^{-5}$, respectively (Eq.
A18).