# Peer review of "Modelling the habitat preference of two key *Sphagnum* species in a poor fen as controlled by"

_Biogeosciences, 2019_

## Referee Comment (RC1) · Anonymous Referee #1 · 8 Dec 2019

1. Does the paper address relevant scientific questions within the scope of BG? YES

2. Does the paper present novel concepts, ideas, tools, or data? YES

3. Are substantial conclusions reached? YES

4. Are the scientific methods and assumptions valid and clearly outlined? MOSTLY YES

5. Are the results sufficient to support the interpretations and conclusions? ALMOST

6. Is the description of experiments and calculations sufficiently complete and precise to allow their reproduction by fellow scientists (traceability of results)? YES

7. Do the authors give proper credit to related work and clearly indicate their own new/original contribution? NO

8. Does the title clearly reflect the contents of the paper? YES

9. Does the abstract provide a concise and complete summary? YES

10. Is the overall presentation well structured and clear? YES

11. Is the language fluent and precise? YES

12. Are mathematical formulae, symbols, abbreviations, and units correctly defined and used? YES

13. Should any parts of the paper (text, formulae, figures, tables) be clarified, reduced, combined, or eliminated? YES

14. Are the number and quality of references appropriate? NO

15. Is the amount and quality of supplementary material appropriate? YES

Dear authors,

Your paper presents an interesting, physiology-based model of competition between two Sphagnum species, attempting a high level of realism in the (large-scale) hydrological regimes and the water-related ecophysiology of the two species studied, S. fallax (lawn species) and S. magellanicum (hummock species). I generally liked the approach, though I have some remarks and questions that I think are worth addressing before publication of the final version.

General comments

1. Please provide a list of abbreviations! It was hard work trying to follow the methods and results without one.

2. A discussion of some literature very relevant to this study, exploring the same ideas though without using a formal model, is missing:

(Titus, et al. 1983, Titus and Wagner 1984). One of the interesting results of these studies is that there is a seasonal dynamic in the water-content response of photo-synthesis. This may be very relevant to your model, if the model is sensitive to these 'water-stress' responses.

(Rydin 1986, 1993a, b, 1997, Rydin and Barber 2001) And more: check the publica-tions by Hakan Rydin, he has been working on competition between Sphagna for a long time.

Another important source, which, however, has not yet been fully published (but a relevant summary with numbers to compare yours against is available in the thesis summary: http://www.diva-portal.org/smash/get/diva2:1282760/FULLTEXT01.pdf ), is the recent PhD thesis by Fia Bengtsson (Uppsala), in particular chapters 4 and 5.

This paper (Hájek 2014) is also very relevant, among other things for some method-ological issues.

3. Model structure: The abstract promises a very wide scope ('dynamic feedback be-tween plant community structure and the environment'), but there is no feedback from the species composition (Modules 1 and 2) on the hydrology (module 3) in the model. Therefore: how does this model really address the feedback you mention? In the discussion, you could also be more explicit about the implications of the species com-position on biogeochemical processes, see e.g. (Bengtsson, et al. 2016, Cornelissen, et al. 2007). Alternatively, do not suggest this focus on feedbacks in the abstract and introduction.

The vertical water transport is implemented in detail, but in the detailed modules 1 and 2 there does not seem to be horizontal water exchange between neighbours, although this may play an important role in maintaining Sf in hummocks, supported by the water held in Sm (Rydin 1985 ; Rydin and McDonald 1985 ; Robroek et al. 2007a ). In your experiments, basing the drying speed on single capitula, the capitulum density, i.e. facilitation between neighbours in retaining water, could not affect the drying speed,

thereby possibly missing part of the difference between the lawn and the hummock species (i.e. under-estimating the difference).

4. Model parameters / results L487 & L520-522 Please also explain why Sf has an advantage over Sm in the lawns. Why does Sf have faster growth? This is not clear to me at all. According to your photosynthesis measurements, Sf has a lower Amax (which seems strange, usually indeed lawn species have higher rates) and the same respiration rate as Sm. Therefore, at high water content and high light, Sm and not Sf should have a benefit in terms of NSC production. As the conversion from NSC to biomass is the same for both species, the only way to explain the higher length growth of Sf in the lawn environment is the higher Hspc (higher height growth per unit biomass). Correct?

5. Ecophysiological measurements / model parameters: L1017 You state here that A tended to increase with time and that it peaked at water contents below the maximum, as indeed shown by the theoretical figure 1B, but not by the measured curves in Fig 2C. Indeed I would have expected such a peak. Can you explain the absence of diffusion limitation in your experiment? Good ventilation..? Is it realistic to measure one capitulum in isolation? Lots of air all around it compared to a capitulum immerged in a (wet) Sphagnum mat... Consequently, also, how homogenously will the capitula have dried out in the GFS compared to in a Sphagnum mat?

It has been shown that the speed of drying during gas exchange measurements can strongly affect the conclusions about optimum water content and water compensation point (Hájek 2014). Under quick drying, as in your experiments, it seems typical to get the type of curves you present. However, under slower drying, as would be typical in the field, the optimum WC would be lower and the depression at high WC stronger. In particular the high compensation point you found, at water contents of up to 600%, seems to be a typical artefact of such fast drying, related to the inhomogenous drying within the capitula.

Also, a field water content of 1470 and 809% water per dry mass seems extremely low for Sphagnum in general and for these species. For S magellanicum I have seen max WC values reported between 2000 and 3000%, and for S. fallax of about 1500% (or 1100%, equivalent to 12 gFM/gDM (Titus, et al. 1983)). You even state yourself (Line 270) that 'it is known that Wmax is around 25-30 g g-1. So I do not understand why you started you experiment at 14,7 and 8,09 g g-1 or where you use these values, as opposed to the values in L277.

If the light curves took up to 120 minutes to complete (why? That is a very long time especially if you only measured at 4 light levels, which seems very little to determine a reliable curve. . .), and drying down to the compensation point took 120-180 minutes, this implies that during the light response measurements you measured a combination of reduced light and reduced water content, so that the curves probably do not reflect only the light response. For determining the Amax this should be no problem, as you started at the highest light level, i.e. at Amax. Are you sure there was no photoinhibition at these high light levels? This may be a problem when starting light response measurements at the high end, as it would affect the rest of the measurements.

6. Model tests: As an important difference between your and previous models lies in the coupling to environmental fluctuations and stochasticity (L97-98), it would make sense to present a test of the importance of these processes to the model output. Would a simpler model provide similarly good results?

I would also be interested in seeing the effects of the water retention and photosynthetic water-response parameters separately. Especially since the parameters for the latter may suffer from some measurement artefacts.

7. Presentation: L279-352 are all about module 3, which seems a bit unbalanced, seeing that modules 1 and 2 seem more important for the competition results. Model 3 is not tested in this paper. . .

Specific comments

L20 In the introduction it could be explained more clearly why a mechanistic model is needed to predict species compositions under changing water levels. Is a prediction based on known habitat preferences not good enough?

L60-61 how does the species composition affect these processes? In particular (for discussion), how do your species / ecological types affect these processes?

L381 it would be interesting to see the effects of water retention and 'water stress' separately

L471 to me it does not look like photosynthesis of S. fallax is more sensitive to changes in the water content, as Amax lies at lower water contents than for S magellanicum, suggesting that it can handle dry conditions better.

L552 how exactly may it serve?

L561 Similarly, how could it be used in DVM development? If you can, please try to be more explicit here.

Table 1: Rs20 was not significantly different between the species, then why use different values here? How large is the effect on the results?

Technical corrections:

L24 employs

L50 why 'during decadal timeframe'?

L57 have

L66 remove 'community'

L69 I do not think that this modelling can be considered a 'space-for-time' approach. The processes are different in space than in time.

L90 . . .that is covered. . ., . . .As competition occurs. . .

L100 within the peatland moss layer

L102 whose competitiveness?

L106 positions a long a

L113 modelled is located

L119 with a sparse cover of vascular plants

L125 The Peatland. . .

L126 explain 'water-energy conditions'

L128 consisting

L132 are driven

L142-143 A is not directly controlled by CWR, please rephrase

L145 These were not really random variables, but variables randomly selected from a distribution

Eq5: what are the rules for the timing of growth? Any relation to WC?

L191 explain where Kimm is based on

L204 ii) biomass, or NSC?

L212 This order of sentences suggests that an exhaustion of NSC storage would be due to lateral growth, which would not make sense, as lateral growth should not take place if NSC supplies are not enough to sustain both new capitula

L217 why suddenly 'moss parameters' - better use the same terms all the time

L227 how does shoot density vary in the model, if you model one capitulum per grid cell?

L235 where is the centre of the moss layer?

[Figure]

L239 what is the 'capacity of water'?

L264 'where Wopt is the optimal water...

L270-278 It is not clear to me why this equation was needed.

L277 Is the same W max used for both species..? An how about the values in Table B1

L294 are listed

L295-313 Why are snow dynamics important for the model?

L318 What are 'periodic lateral boundary conditions'?

L323 of the model

L346-347 WTs is the multi-year mean of weekly water table?

L474 insert return

L487 This would be a good place to explain why Sf overgrowsn Sm in the lawns.

L495 in other hydraulic

L513 Explain the 'this could be because', this is not obvious

L520 As Amax was lower in Sf, and Rs20 was the same, it seems that only Hspec would explain the result. You could repeat the test adjusting only Hspec to test this.

L527 dominated

L544 This would be a good place to explain how these impacts work and what your model thus implies (or could imply when tested under climate-change conditions) for peatland stability and functioning

Table 1: I would recommend adding the units inside the table

Table 1 & Table B1: A in bryophytes is usually expressed in nmol $g^{-1}$ $s^{-1}$, to avoid to many 0 before significant digits start.

Table 2 and 3: please explain abbreviations

Appendix L 150 at one hertz?

L209 The software is R, R Studio is just an interface

Fig B2: it is impossible to distinguish the models form the data especially in C. See comments above about the curves in C.

Cited references:

Bengtsson, F., Granath, G. and Rydin, H. 2016. Photosynthesis, growth, and decay traits in Sphagnum - a multispecies comparison. - Ecology and evolution 6: 3325-41.

Cornelissen, J. H. C., Lang, S. I., Soudzilovskaia, N. A. and During, H. J. 2007. Comparative cryptogam ecology: A review of bryophyte and lichen traits that drive biogeochemistry. - Annals of Botany 99: 987-1001.

Hájek, T. 2014. Physiological ecology of peatland bryophytes. - In: Hanson, D. T. and Rice, S. K. (eds), Photosynthesis in early land plants. Springer, pp. 233–252.

Rydin, H. 1986. Competition and niche separation in Sphagnum. - Can J Bot 64: 1817-1824.

Rydin, H. 1993a. Interspecific competition between Sphagnum mosses on a raised bog. - Oikos 66: 413-423.

Rydin, H. 1993b. Mechanisms of interactions among Sphagnum species along water level gradients. - Advances in Bryology 5: 153-185.

Rydin, H. 1997. Competition among bryophytes. - Advances in Bryology 6: 135-268.

Rydin, H. and Barber, K. E. 2001. Long-term and fine-scale coexistence of closely related species. - Folia Geobot 36: 53-61.

Titus, J. E., Wagner, D. J. and Stephens, M. D. 1983. Contrasting Water Relations of Photosynthesis for 2 Sphagnum Mosses. - Ecology 64: 1109-1115.

Titus, J. E. and Wagner, D. J. 1984. Carbon Balance for 2 Sphagnum Mosses - Water-Balance Resolves a Physiological Paradox. - Ecology 65: 1765-1774.

---

## Referee Comment (RC2) · Anonymous Referee #3 · 6 Feb 2020

General Comments In this manuscript, the authors develop and validate a simulation model (Peatland Moss Simulator, PMS) that combines an underlying (and previously published) peatland hydrology model with individual-based Sphagnum surface physiology and community components (unpublished) to be used in future studies projecting peatland response to environmental change. They suggest that the PMS will better capture the "feedback between the plant community structure and the environment" that is lacking in other models. In my opinion, this is a very worthwhile approach at the present as there is much known about each of the components, and their degree of variation, that they incorporate into their model; the field is ripe for such a dynamic, quantitative summary of community and ecosystem processes in Sphagnum domi-

nated peatlands.

Overall, this is a well-written and well organized manuscript. The main modeling approach is clearly laid out in Fig 1 and the components well described and justified for each. My expertise lies in the area of physiology/ecology and I read those sections with particular interest. The relationships between capitulum water and carbon dynamics were thoughtfully approached. Indeed, their ability to highlight the importance of water balance parameters relative to ones that affect growth is an important result (L507-525). Also, I appreciated their using a carbon allocation model that segregated non-structure carbon (NSC) as the pool for new growth. This level of physiological mechanism of often left out. In general, I found the level of their modeling suitably mechanistic. This also applied to the competition model where an individual approach tied to growth and competitive interactions seems appropriate, especially given the well-developed state of knowledge in this area.

I do like the modeling within each of the surface components. However, the strength of the manuscript is linking them to an underlying hydrology model, which provides a sound ecological context. In addition, their tests against predictions and field data seem appropriate.

That said, in my opinion, there are areas that could be improved. The major and minor ones I list below with less important ones indicated in the Specific Comments section. Major Comments A. The Abstract and Introduction focus on feedbacks between the plant community structure and the environment. It seems from the outline of the model (Fig 1) and the descriptions of it that the environment serves as more of a forcing variable on the plant physiology and community dynamics. For example, there are no processes that feedback to the "Community environment" module in their model (Fig 1) and I did not see any not listed within the descriptions of the model structure in the text. Clearly there are feedbacks between the capitula environment module and the shoot growth and competition module, but I don't think the capitula environment is really what people would consider part of the plant's environment. Fixing this will

reframe the justification, but I think it can still be well justified. B. In my opinion, the paper would be improved by applying the model to make predictions about a particular response to an environmental change. It could be argued that this paper is for model development and validation and the next one will use it in a predictive context. However, is there a small question that could be addressed with the model that would illustrate its value? C. I was surprised that the model did not deal with any of the autogenic processes that lead to hummock formation. The community model is spatially explicit and it would seem that it would allow for rule-based hummock formation simulation when succeeding from a high water table. Instead, the model simulates either high or low water tables. This seems like hummock forming processes would represent a true feedback to the environment. Is this either desirable or possible in this model iteration? D. The living tissue of Sphagnum species clearly differ in their hydraulic conductivity (Km, p8; as shown in the McCarter and Price 2014 paper cited, see also Li, Glime and Liao 1992, J Bryology 17:59); however, this was treated as a constant. Although I do not think there are reports of how this differs between S. magellanicum and S. fallax, I think it would be important to consider variation in this using hummock and hollow values for the two. I suspect that this would only accentuate the differences they observe in their results, and/or, speed up the time until species distributions equilibrate. In any case, given that species cover changes are quite sensitive to Km (Table 3), I think it is worth modeling species-specific differences in this parameter.

Minor Comments E. I was surprised that the Titus and Wagner (1984, Ecology 65:1765) paper was not cited. Some of the simulation modeling is similar and should make for a nice comparison. F. Need a table of abbreviations. G. It would be very helpful to show how the parameter values used fall within reported ranges in the literature (e.g., Table 1).

Specific Comments 1. Line 81-2 Aren't they linked by capitulum water balance? Retention is too specific, I think. 2. L101-4 I find this sentence confusing. Can you be more clear about the linkages? 3. L142-3 I think it is controlled by water content—not

the same thing as water retention. 4. Fig 1. What is the difference between dashed and solid lines? Can the boxes or arrows be changed so it is easier to tell that Module III influences Module II—it took a while to realize it wasn't controlled by precip and evap, where I thought the arrows were coming from. I would suggest making the figure legend more complete. 5. L213-18 This is the discussion of reseeding. It would be useful to know how frequently this was necessary. Was it rare with little impact on results or more common? 6. L380-82 Is it worth listing what the parameters are meant here in text as is done below? 7. Fig 2. The y-axis for the top figure should be "Relative Cover". Also, can you use solid and dashed lines to distinguish Hummock from Lawn? Would make it easier to read on B&W print. 8. L415 Why not show both species in both environments? Here only show S. mag in hummocks and S. fal in lawns. 9. L418-23 Would it be better to report these as elasticities (% change in outcome per % change in parameter). This is easy to do as they were all set to vary by +/- 10%. However, this would allow you to assess whether or not it was a large change or not—what would cutoff be? You report that being less than the standard deviation for a 10% change is meaningful (L490), can you defend that? 10. L469 You state that S. fallax capitula were less resistant to evaporation, but the data in Table B1 seem to indicate otherwise (see ra; this result is opposite to what I would expect although the do not differ significantly). 11. L492 Yes, it would be expected for n to have a large effect as it is a scaling factor, so changes in its magnitude get amplified. 12. L502-06 See Comment D above. 13. L968 The procedure for doing the photosynthetic measurements would seem to cause quite a lot of drying within the cuvette (RH 60%, impeller at level 5) where they were measured over 60-120 minutes. Were they rewetted following each light level? Were they allowed to dry? How did mass change during the course of the measurement and did that influence shape of curve? Can you provide a light response curve showing data? If there are not good answers to these questions, it would at least be helpful to include how the parameters measured compare with other ones in the literature.

---

## Author Comment (AC1) · 30 Mar 2020

We are grateful to two anonymous reviewers that had put a lot of effort to improve our manuscript. Accordingly, we did our best to follow the suggestions. In those few cases where we disagreed or were not able to do that, we explain why. Please find our responses to each comment below.

Referee 1

General comments

1. Please provide a list of abbreviations! It was hard work trying to follow the methods

and results without one.

R: We now provide a list of symbols and abbreviations as suggested (new Table 1).

2. A discussion of some literature very relevant to this study, exploring the same ideas though without using a formal model, is missing: (Titus, et al. 1983, Titus and Wagner 1984). One of the interesting results of these studies is that there is a seasonal dynamic in the water-content response of photosynthesis. This may be very relevant to your model, if the model is sensitive to these 0water-stress0 responses. (Rydin 1986, 1993a, b, 1997, Rydin and Barber 2001) And more: check the publications by Hakan Rydin, he has been working on competition between Sphagna for along time.

Another important source, which, however, has not yet been fully published (but a relevant summary with numbers to compare yours against is available in the thesis summary: http://www.diva-portal.org/smash/get/diva2:1282760/FULLTEXT01.pdf ), is the recent PhD thesis by Fia Bengtsson (Uppsala), in particular chapters 4 and 5. This paper (Hájek 2014) is also very relevant, among other things for some methodological issues.

R: Thank you for pointing out missing references to relevant literature. Indeed we were missing quite a number of classics and new ones that are now used to deepen Intro and Discussion. Originally, we presented model development and empirical measurements in two separate manuscripts; In the merging we had accidentally lost a big part of references but now they are included again.

3. Model structure: The abstract promises a very wide scope (0dynamic feedback between plant community structure and the environment0), but there is no feedback from the species composition (Modules 1 and 2) on the hydrology (module 3) in the model. Therefore: how does this model really address the feedback you mention?

R: Our model lacks the feedback to hydrology as the referee pointed out. We now removed the parts of Abstract and Introduction that give reader a reason to expect

otherwise.

In the discussion, you could also be more explicit about the implications of the species composition on biogeochemical processes, see e.g. (Bengtsson, et al. 2016, Cornelissen, et al. 2007). Alternatively, do not suggest this focus on feedbacks in the abstract and introduction.

R: In the discussion we now describe the implications of the species composition on biogeochemical processes via their traits.

The vertical water transport is implemented in detail, but in the detailed modules 1 and 2 there does not seem to be horizontal water exchange between neighbours, although this may play an important role in maintaining Sf in hummocks, supported by the water held in Sm (Rydin 1985 ; Rydin and McDonald 1985 ; Robroek et al. 2007a ). In your experiments, basing the drying speed on single capitula, the capitulum density, i.e. facilitation between neighbours in retaining water, could not affect the drying speed, thereby possibly missing part of the difference between the lawn and the hummock species (i.e. under-estimating the difference).

R: Our model also lacks horizontal water transport that has found to allow individuals of lawn species to be present in dried habitats. The pattern is interesting and may play a role in speeding up the spreading of lawn species when conditions become wetter. Unfortunately, in this first attempt to mechanistically model Sphagnum community dynamics we were only able focus getting the general distribution pattern realistic and leave perfection for later. In this stage essential data for parameter values not yet exist for quantifying horizontal water transport among neighboring individuals such as hydraulic conductivity. The model can be improved further when the parameterization could be supported by experimental studies.

In our drying experiment a layer of capitula, with same density as in field was placed on the cuvette, therefor the neighbours do to some extend affect the drying process, yet, the stems are lacking and it surely does not truly reflect the field conditions.

Speed as such was not yet our focus but the response of photosynthesis to water content, and we do think our approach catches the between species differences in this process.

4. Model parameters / results L487 & L520-522 Please also explain why Sf has an advantage over Sm in the lawns. Why does Sf have faster growth? This is not clear to me at all. According to your photosynthesis measurements, Sf has a lower Amax (which seems strange, usually indeed lawn species have higher rates) and the same respiration rate as Sm. Therefore, at high water content and high light, Sm and not Sf should have a benefit in terms of NSC production. As the conversion from NSC to biomass is the same for both species, the only way to explain the higher length growth of Sf in the lawn environment is the higher Hspc (higher height growth per unit biomass). Correct?

R: The explanation suggested by the Refree 1 is correct. We have now written out that the bigger height growth of S. fallax per biomass production rate is because of its looser structure. Like us, Bengtson et al. (2016) measured similar photosynthesis rate for the two species, but clearly higher height growth for S. fallax. (Bengtsson, F., Granath, G., & Rydin, H. (2016). Photosynthesis, growth, and decay traits in Sphagnum – a multispecies comparison. Ecology and Evolution, 6(10), 3325-3341.)

5. Ecophysiological measurements / model parameters: L1017 You state here that A tended to increase with time and that it peaked at water contents below the maximum, as indeed shown by the theoretical figure 1B, but not by the measured curves in Fig 2C. Indeed I would have expected such a peak. Can you explain the absence of diffusion limitation in your experiment? Good ventilation..? Is it realistic to measure one capitulum in isolation? Lots of air all around it compared to a capitulum immerged in a (wet) Sphagnum mat: : : Consequently, also, how homogenously will the capitula have dried out in the GFS compared to in a Sphagnum mat?

R: The expected peak was actually there, see redrawn figure B2C. For some reason (not clear to us anymore) we had earlier cut the X-axis (capitulum water content) shorter in panel C than in the other panels.

We did not measure single capitulum in isolation but a layer of capitula was placed on cuvette (see Fig. B1A). We rewrote the related methods section to make them clearer.

It has been shown that the speed of drying during gas exchange measurements can strongly affect the conclusions about optimum water content and water compensation point (Hájek 2014). Under quick drying, as in your experiments, it seems typical to get the type of curves you present. However, under slower drying, as would be typical in the field, the optimum WC would be lower and the depression at high WC stronger. In particular the high compensation point you found, at water contents of up to 600%, seems to be a typical artefact of such fast drying, related to the inhomogenous drying within the capitula.

R: In slow drying (Hájek 2014), environmental vapor pressure remains constant and evaportation rate decreases with time. In such experimental conditions water movement could be sufficiently rebalanced between internal and external tissues, so that the water potential becomes equilibrized among different parts of capitulum. However, in field conditions, evaporation demand could be more strongly driven by radiation than vapor pressure deficit, particularly during a hot clear summer day. Thus, it could be much faster than in a dessication chamber and consequently, the water content may not rebalance fast enough to reach equalibrium. Moreover, the branch leaves in the outer part of capitula could be more photosynthetically active than the internal core parts. As the drying is heterogenous, photosynthesis rate could be largely reduced just by the drying of outer tissues, even though the internal core part could be wetter. This is also supported by our measurement, which showed a higher compensation point for photosynthesis than that from the slow drying experiment (Hájek 2014). Therefore, we believe the fast drying could be a better imitation of field processes.

Also, a field water content of 1470 and 809% water per dry mass seems extremely

low for Sphagnum in general and for these species. For S magellanicum I have seen max WC values reported between 2000 and 3000%, and for S. fallax of about 1500% (or 1100%, equivalent to 12 gFM/gDM (Titus, et al. 1983)). You even state yourself (Line270) that it is known that Wmax is around 25-30 g g-1. So I do not understand why you started you experiment at 14,7 and 8,09 g g-1 or where you use these values, as opposed to the values in L277.

R: The reason for the low field water contents compared to earlier published values lies in the measurement method we used (as explained in supplementary material). We measured the capitulum and stem section WC separately and allowed the external water on Sphagnum surfaces to dry out before weighing the fresh weight. We started the experiment on the water content levels where excess water does not limit photosynthesis. This optimal WC is now shown in redrawn Figure B2C, which now starts already in a higher water content. We have now tried to explain this better in Methods.

If the light curves took up to 120 minutes to complete (why? That is a very long time especially if you only measured at 4 light levels, which seems very little to determine a reliable curve: : :), and drying down to the compensation point took 120-180 minutes, this implies that during the light response measurements you measured a combination of reduced light and reduced water content, so that the curves probably do not reflect only the light response. For determining the Amax this should be no problem, as you started at the highest light level, i.e. at Amax. Are you sure there was no photoinhibition at these high light levels? This may be a problem when starting light response measurements at the high end, as it would affect the rest of the measurements.

R: It is true that the light response curve cannot exclude the impact of drying. To mitigate the impact, we have measured the photosynthesis at highest light level from the beginning of each measurement, then decreased the light level sequentially (as respiration could be less sensitive to drying).

We have added more details on the measurement protocol and choice of light levels.

The cuvette relative humidity was kept at 80% to slow down the drying process, but not to cause damage to the devise. The maximum light level 1500 PPFD was chosen based on our earlier studies with more light levels (Laine 2011, 2015) where we had not observed any photoinhibition until PPFD 2000, and A were often still increasing between PPFD 800 and 1500. Laine, A. M., Juurola, E., Hájek, T., & Tuittila, E. S.: Sphagnum growth and ecophysiology during mire succession. Oecologia, 167(4), 1115-1125, 2011. Laine, A. M., Ehonen, S., Juurola, E., Mehtätalo, L., & Tuittila, E. S.: Performance of late succession species along a chronosequence: Environment does not exclude Sphagnum fuscum from the early stages of mire development. Journal of vegetation science, 26(2), 291-301, 2015.

6. Model tests: As an important difference between your and previous models lies in the coupling to environmental fluctuations and stochasticity (L97-98), it would make sense to present a test of the importance of these processes to the model output. Would a simpler model provide similarly good results?

R: We believe that the main purpose of modelling is to illustrate the reality and serve as a tool for systematic assessment of the processes. Simple community models without individual-based processes implicitly weigh on generality and forgive outliers. However, environmental fluctuation and extremes are becoming more frequent and intensive with climate change, and this is likely to give advantage to an otherwise unlikely change in peatland community. To help with this situation, our modelling is able to populate outputs along a probability distribution and allows assessing individuals with different trait combinations as a part of the probabilities. As these models are fundamentally different in focuses and underlying mechanisms, simply comparing the goodness of results seems pointless.

I would also be interested in seeing the effects of the water retention and photosynthetic water-response parameters separately. Especially since the parameters for the latter may suffer from some measurement artefacts.

R: This is a very appreciated comment. Our future goal is also to make the picture clearer and understanding the factorial effects is a very important aspect. At the moment, our data and techniques are insufficient to separate the different effects. Therefore, model testing based on the parameters quantified by the "mixed" information could be less informative, unless we have had improved measurement data.

In addition, S. fallax and S. magellanicum are largely different in both water retention and photosynthetic response to water stress. Further testing on species either with similar water retention, or with similar photosynthetic response would be more informative to this question.

7. Presentation: L279-352 are all about module 3, which seems a bit unbalanced, seeing that modules 1 and 2 seem more important for the competition results. Model 3 is not tested in this paper:

R: Module 3 is about environment and it was not tested here because it was not in the focus of this paper. However, to bridge environmental fluctuation to community processed, our center of the focus, we needed to set up the environment first.

Specific comments

L20 In the introduction it could be explained more clearly why a mechanistic model I needed to predict species compositions under changing water levels. Is a prediction based on known habitat preferences not good enough?

R: The species known preference along the prevailing moisture gradient might not directly serve as a reliable predictor for future species compositions as water table fluctuation is likely to increase. This is now added in Introduction.

L60-61 how does the species composition affect these processes? In particular (for discussion), how do your species / ecological types affect these processes?

R: through interspecific variability in species traits such as photosynthetic potential and litter quality
L381 it would be interesting to see the effects of water retention and water stress separately

R: See above the response to 6

L471 to me it does not look like photosynthesis of S. fallax is more sensitive to changes in the water content, as Amax lies at lower water contents than for S magellanicum, suggesting that it can handle dry conditions better.

R: In this study, we use the term sensitivity to represent the dependency of photosynthesis changes to water content changes in capitula. Although S. fallax has greater tolerance to relatively low water content, the water content change for photosynthesis to drop from maximum to zero was much smaller than S. magellanicum (B2C). This is why we claim that photosynthesis of S. fallax is more sensitive to changes in the water content. This is now better pointed out in the text.

L552 how exactly may it serve?

R: we have removed the sentence

L561 Similarly, how could it be used in DVM development? If you can, please try to be more explicit here.

R: We introduced a mechanism to include competition based on growth rates that could be used in building dynamic community structure into DVMs.

Table 1: Rs20 was not significantly different between the species, then why use different values here? How large is the effect on the results?

R: These values are measured from field experiments and reported here. Although the means are not significantly different, we cannot judge that the probability distributions are the same, based on only several samples. Therefore, we used the measured means and standard deviations to generate probability distributions for each species.

Technical corrections:

L24 employs

R: corrected

L50 why "during decadal timeframe"?

R: not within few years but faster than a hundred yeas

L57 have

R: corrected

L66 remove "community"

R: removed

L69 I do not think that this modelling can be considered a "space-for-time" approach. The processes are different in space than in time.

R: removed

L90 : : :that is covered: : :, : : :As competition occurs: : :

R: modified as suggested

L100 within the peatland moss layer

R: added

L102 whose competitiveness?

R: clarified

L106 positions a long a

R: corrected

L113 modelled is located

R: modified as suggested

L119 with a sparse cover of vascular plants

R: modified as suggested

L125 The Peatland: : :

R: added

L126 explain "water-energy conditions"

R: clarified

L128 consisting

R: modified

L132 are driven

R: modified

L142-143 A is not directly controlled by CWR, please rephrase

R: rephrased

L145 These were not really random variables, but variables randomly selected from a distribution

R: corrected

Eq5: what are the rules for the timing of growth? Any relation to WC?

R: Timing of growth is controlled by a temperature threshold and NSC availability. Growth occurs when T > 5 °C and NSC is above zero. The dynamics of NSC storage is related to WC through net photosynthesis.

L191 explain where Kimm is based on

R: Reference added to Asaeda, T. and Karunaratne (2000)

Asaeda, T. and Karunaratne, S.: Dynamic modelling of the growth of Phragmites australis: model description, Aquatic Botany, 67, 301-318, 2000.

L204 ii) biomass, or NSC?

R: NSC; corrected

L212 This order of sentences suggests that an exhaustion of NSC storage would be due to lateral growth, which would not make sense, as lateral growth should not take place if NSC supplies are not enough to sustain both new capitula

R: Indeed, it does not make sense. Removed

L217 why suddenly "moss parameters' - better use the same terms all the time

R: reformulated

L227 how does shoot density vary in the model, if you model one capitulum per grid cell?

R: Ds is BM per grid cell, not the number of capitula. The (suggested) table of abbreviations with their units will clary this.

L235 where is the centre of the moss layer?

R: removed

L239 what is the 0capacity of water0?

R: corrected to "water uptake capacity"

L264 0where Wopt is the optimal water: : :

R: reformulated

L270-278 It is not clear to me why this equation was needed.

R: In Eq. 11 we evaluated the water stress effect at high Wcap conditions, which are

beyond the upper boundary of our drying experiment. Therefore in Eq. 12 we used a brief method to estimate the capitula Wcap from volumetric water content of moss carpet.

L277 Is the same W max used for both species..? An how about the values in Table B1

R: Yes, same value is used for both species. This is a theoretical maximum for high water-content restrictions on photosynthesis (Frolking et al., 2002), which is needed but not our focus in the modelling.

L294 are listed

R: changed

L295-313 Why are snow dynamics important for the model?

R: Snow dynamics impact environmental conditions in the early growing season. As they are currently under change due to climate change, we considered important to include them for better predictions.

L318 What are "periodic lateral boundary conditions"?

R: rewritten

L323 of the model

R: added

L346-347 WTs is the multi-year mean of weekly water table?

R: clarified

L474 insert return

R: I was not able to find were to insert

L487 This would be a good place to explain why Sf overgrowsn Sm in the lawns.

R: Explanation included. Basically, the looser structure of S. fallax allows its faster height growth.

L495 in other hydraulic R: added

L513 Explain the 0this could be because0, this is not obvious

R: the text was quite unclear, now clarified

L520 As Amax was lower in Sf, and Rs20 was the same, it seems that only Hspec would explain the result. You could repeat the test adjusting only Hspec to test this.

R: Hspec is a very powerful trait but our focus here was not to discuss each trait. Also, we don't have a species that would have lower in Hspec but resembles S. fallax in other traits. Therefore, we don't understand why this test would be meaningful.

L527 dominated

R: modified

L544 This would be a good place to explain how these impacts work and what your model thus implies (or could imply when tested under climate-change conditions) for peatland stability and functioning

R: Explained

Table 1: I would recommend adding the units inside the table

R: added

Table 1 & Table B1: A in bryophytes is usually expressed in nmol g-1 s-1, to avoid to many 0 before significant digits start.

R: we prefer to use the current version

Table 2 and 3: please explain abbreviations

R: explanation added

Appendix L 150 at one hertz?

R: Changed to every second.

L209 The software is R, R Studio is just an interface

R: corrected

Fig B2: it is impossible to distinguish the models form the data especially in C. See comments above about the curves in C.

R: The lines have now been redrawn. Fig B2 shows only measured values.

---

## Author Comment (AC2) · 30 Mar 2020

Major Comments

A. The Abstract and Introduction focus on feedbacks between the plant community structure and the environment. It seems from the outline of the model (Fig 1) and the descriptions of it that the environment serves as more of a forcing variable on the plant physiology and community dynamics. For example, there are no processes that feedback to the "Community environment" module in their model (Fig 1) and I did not see any not listed within the descriptions of the model structure in the text. Clearly there are feedbacks between the capitula environment module and the shoot growth

and competition module, but I don't think the capitula environment is really what people would consider part of the plant's environment. Fixing this will reframe the justification, but I think it can still be well justified.

R: The bold mentions on the feedback to hydrology in Abstract and Introduction are now removed as they were misleading.

B. In my opinion, the paper would be improved by applying the model to make predictions about a particular response to an environmental change. It could be argued that this paper is for model development and validation and the next one will use it in a predictive context. However, is there a small question that could be addressed with the model that would illustrate its value?

R: We agree that applying the model to predict change in community structure as a response to environmental change would be a logical next step and make the story far more interesting. However, as we are already here combining new empirical measurements conducted for model parameterizing and testing and description of the new model (and ending having a lot of text, tables and figures as appendix to keep the story readable) we see that adding more would be just too much.

C. I was surprised that the model did not deal with any of the autogenic processes that lead to hummock formation. The community model is spatially explicit and it would seem that it would allow for rule-based hummock formation simulation when succeeding from a high water table. Instead, the model simulates either high or low water tables. This seems like hummock forming processes would represent a true feedback to the environment. Is this either desirable or possible in this model iteration?

R: We agree that our model will be an excellent starting point to address autogenic processes that lead to hummock formation by including feedback to hydrology. We see PMS, the first model addressing Sphagnum community dynamics, as a steppingstone for the future work in numerical conceptualizing of peatland processes.

D. The living tissue of Sphagnum species clearly differ in their hydraulic conductivity (Km, p8; as shown in the McCarter and Price 2014 paper cited, see also Li, Glime and Liao 1992, J Bryology 17:59); however, this was treated as a constant. Although I do not think there are reports of how this differs between S. magellanicum and S. fallax, I think it would be important to consider variation in this using hummock and hollow values for the two. I suspect that this would only accentuate the differences they observe in their results, and/or, speed up the time until species distributions equilibrate. In any case, given that species cover changes are quite sensitive to Km (Table 3), I think it is worth modeling species-specific differences in this parameter.

R: We agree, but species-specific data on the hydraulic conductance was generally lacking. It would be very intriguing to see the impact of these parameters on modelling results, once the measurement date becomes available.

Minor Comments

E. I was surprised that the Titus and Wagner (1984, Ecology 65:1765) paper was not cited. Some of the simulation modeling is similar and should make for a nice comparison.

R: Now included

F. Need a table of abbreviations.

R: Added

G. It would be very helpful to show how the parameter values used fall within reported ranges in the literature (e.g., Table 1).

R: We did a large search to fulfil this. Although we were able to find some meaningful values for comparison in the literature, we did not find them for most of the parameters and many of the ones related to photosynthesis were not measured in comparable conditions. For these reasons we abandoned the good idea to include ranges in the table but took the few ones found as subnotes in Table2 (previous Table 1).

Specific Comments

1. Line 81-2 Aren't they linked by capitulum water balance? Retention is too specific, I think. R: modified

2. L101-4 I find this sentence confusing. Can you be more clear about the linkages? R: Rewritten to be clearer, as suggested by both reviewers.

3. L142-3 I think it is controlled by water content âËŸAËĞ Tnot the same thing as water retention. R: Rewritten

4. Fig 1. What is the difference between dashed and solid lines? Can the boxes or arrows be changed so it is easier to tell that Module III influences Module IIâËŸA ËĞ Tit took a while to realize it wasn't controlled by precip and evap, where I thought the arrows were coming from. I would suggest making the figure legend more complete.

R: In revised Fig 1, we added instructions to submodule boxes, replaced arrows from water balance to evaporation and capillary flow and added legend for different types of arrows in the figure.

5. L213-18 This is the discussion of reseeding. It would be useful to know how frequently this was necessary. Was it rare with little impact on results or more common?

R: The re-establishment from spore is calculated annually but it was not common in general. Most changes in grid cell occupants come from the invasion from neighboring cells. This process was mostly observed in the first two years of simulation, as the trait combination were randomly chosen, and consequently some combinations performed too poor to support the survival of individuals.

6. L380-82 Is it worth listing what the parameters are meant here in text as is done below?

R: We added list of symbols and abbreviations (New Table 1)

7. Fig 2. The y-axis for the top figure should be "Relative Cover". Also, can you use

solid and dashed lines to distinguish Hummock from Lawn? Would make it easier to read on B&W print. changed as suggested

R: Modified as suggested

8. L415 Why not show both species in both environments? Here only show S. mag in hummocks and S. fal in lawns. R: We have empirical data only from their natural habitats

9. L418-23 Would it be better to report these as elasticities (% change in outcome per % change in parameter). This is easy to do as they were all set to vary by +/- 10%. However, this would allow you to assess whether or not it was a large change or notâËŸA ËĞTwhat would cutoff be? You report that being less than the standard deviation for a 10% change is meaningful (L490), can you defend that?

R: We appreciate this suggestion and modified our statement

10. L469 You state that S. fallax capitula were less resistant to evaporation, but the data in Table B1 seem to indicate otherwise (see ra; this result is opposite to what I would expect although the do not differ significantly).

R: Rewritten to clarify. This was obviously unclearly expressed as it confused both reviewers.

11. L492 Yes, it would be expected for n to have a large effect as it is a scaling factor, so changes in its magnitude get amplified.

R: added to the text

12. L502-06 See Comment D above.

R: see response to D

13. L968 The procedure for doing the photosynthetic measurements would seem to cause quite a lot of drying within the cuvette (RH 60%, impeller at level 5) where they

were measured over 60-120 minutes. Were they rewetted following each light level? Were they allowed to dry? How did mass change during the course of the measurement and did that influence shape of curve? Can you provide a light response curve showing data? If there are not good answers to these questions, it would at least be helpful to include how the parameters measured compare with other ones in the literature.

R: We have added more details on the measurement protocol. The cuvette relative humidity was kept at 80% to slow down the drying process, but not to cause damage to the devise.

---

## Author Response (AR2)

| 1 | Response letter | 26th june 2020 |

Response letter                                                    26th june 2020
We highly value the comments and suggestions of Reviewers and we have improved the
manuscript accordingly. We hope you find the answers and the revised version appropriate.
Here below are our answers to the comments and manuscript with changes
**Associate Editor Decision: Publish subject to minor revisions (review by editor)** (04 Jun
2020) by Michael Bahn
Comments to the Author:
Dear authors,
both reviewers think that their concerns have been largely addressed, even though some of their
suggestions for additional relevant testing were not taken up by you. The reviewers have
provided a number of excellent specific suggestions for further improvement, which I kindly ask
you to consider during this next round of revisions. Please include also a more critical
assessment of the limitations of the current model version.
Best regards,
Michael Bahn
*-We have now added a critical assessment of the limitations of the current model version to the*
*end of the discussion*
Referee 1

This is my second review of this manuscript for Biogeosciences and I am pleased that the authors incorporated many of the suggestions made by me and the other reviewer. In brief, the strength of the study emerges from their combining an individual-based, two species competition model of Sphagnum community interactions with underlying functional models of water and carbon dynamics and linking it to a hydrology simulation model. Historically, most modeling efforts have focused on understanding and modeling the mechanisms of competition and function in Sphagnum. One of the highlights of this study is the connection to an existing hydrology and surface exchange model that was used to generate local environmental conditions that served as forcing variables for the simulation. Overall, I believe that this manuscript represents a valuable step in developing predictive models of peatland function.

I still have a couple of issues that I think the authors ought to address and then provide a list of minor edits.

A. There are still many places where "water retention" and "water content" are used synonymously. In my mind, retention is when existing water is not released via evaporation or drainage. I don't think this is the meaning the authors intend. Most of the times, I think the authors mean capitulum "water content". This incorporates fluxes in and out, but also the capacity for storage. I think the language needs to be clarified and it begins with its usage in the title. Here is a list of places that I found places where this should be clarified: L2, L28, L35, L75, and L522; there may be others.

*-Changed as suggested.*

B. I remain surprised that the authors still did not show sample light response curves in the Appendix, which were the source of many of their photosynthetic parameters; both reviewers questioned their measurements, especially with regards to the long time for samples to desiccate within the chamber. In the appendix, data for the photosynthetic—water content relationships are shown (Fig B2). I think light response curves should be added. They may be very useful if this study is used for comparative purposes in the future.

*-The measured light response curves for S. magellanicum and S. fallax are now added to Appendix B as Figure B1.*
Minor edits
C. L15: what do you mean by "dynamic community structure"? There should be a better way to state this.
*-Clarified in lines 15-19 to "Current peatland models generally treat vegetation as a static community, although plant community structure is known to alter as a response to environmental change. Because vegetation structure and ecosystem functioning are tightly linked, realistic projections of peatland response to climate change requires including vegetation dynamics in ecosystem models."*

D. L25. The description of what PMS does should be more clear. Perhaps it would be better worded by placing competition up front. "PMS employs a, stochastic, individual-based approach to simulating competition based on species' differences in functional traits."

*Changed in lines 25-30 into: In this study, we developed the Peatland Moss Simulator (PMS), simulating community dynamics of the peatland moss layer. PMS is a process-based model that employs a stochastic, individual-based approach simulating competition within peatland moss layer based on species differences in functional traits.*

E. L35: replace "retention" with "relations"
*-changed*

F. L53: unclear what "dynamic community structure" relates to. "lack mechanisms that underlie and cause dynamic community processes"?

*-Clarified. Now says "Peatland models have generally considered vegetation structure unrealistically as static component"*

G. L55: delete "in order"

*-deleted*

H. L55: delete after "and the research community"
*-deleted*

I. L73: replace "the peers" with "its peers".
*-changed*

J. L75: and water storage
*-added*

K. L87: I think the ability to carry out such a quantitative study is not new and don't believe this has been a great hindrance.
*-deleted*

L. L93: occur to occurs
*-changed*

M. L98: rely to relies
*-changed*

N. L114: replace "locates" to "is located"-*changed*
O. L118: to "10% of the surface is occupied…."
*-changed*

P. L128: to "flow with…"
*-changed*

Q. L143: "race for space" to competition
*-changed*

R. L414: change PCS to PMS
*-changed*

S. L491: lawnss to lawns
*-changed*
T. L535: delete the
*-deleted*

U. L573 and 577: replace dynamical with dynamic
*-changed*

Referee 2

Dear authors,

Thank you for addressing the concerns raised by me and the other reviewer. Several points have now been clarified in the text (please do check the grammar, which is not flawless in the added sections). There are a few remarks that you have responded to only in your reply to us but not in the main text (e.g. my previous points 3b (horizontal water exchange), point C and D of reviewer 3 (autogenic processes leading to hummock formation and differences in moss hydraulic conductivity)). I think those are missed chances of improving the outlook section for your model. In general, I would appreciate a more explicit acknowledgment of the limitations of your model. After all, even if you managed to recreate some ´realistic´ patterns, several potentially important processes are still missing from the model, so you cannot be sure whether you produced these patterns for the right reasons. It is absolutely fine to start with a simple model (even if your main purpose is to `illustrate the reality´ L188 in your response) and to ´leave perfection for later´ (L70 in your response), but thereby it is helpful to line out the path to perfection (well, at least to a model in which the importance of additional processes has been tested) for the readers.

*-We have now included a section in the discussion that focus on the limitation of our model, in lines 588-617. We had that in the first version of the manuscript but is was accidentally left out when we combined the two manuscripts (empirical and modelling) into one and tried to make it concise.*

Also, I think there are two points (6 and 7) that I think you may have misunderstood, so that I will try to formulate them better here. I pasted my old remark and your reply in here to keep track of the context.

My previous point 6 and your reply:

As an important difference between your and previous models lies in the coupling to environmental fluctuations and stochasticity (L97-98), it would make sense to present a test of the importance of these processes to the model output. Would a simpler model provide similarly good results?

R: We believe that the main purpose of modelling is to illustrate the reality and serve as a tool for systematic assessment of the processes. Simple community models without individual-based processes implicitly weigh on generality and forgive outliers. However, environmental fluctuation and extremes are becoming more frequent and intensive with climate change, and this is likely to give advantage to an otherwise unlikely change in peatland community. To help with this situation, our modelling is able to populate outputs along a probability distribution and allows assessing individuals with different trait combinations as a part of the probabilities. As these models are fundamentally different in focuses and underlying mechanisms, simply comparing the goodness of results seems pointless.

My new comment: In this case I was not suggesting that you compare your model to previous, unrelated, models, but that you do tests with your own model, simplifying some processes and seeing if that degrades the results. E.g. instead of using realistic parameter distributions just use a random number generator to e.g. modify the length growth of individual shoots (grid cells). Instead of using realistic environmental fluctuations just use the smoothed mean monthly climate. These are just some examples, I am sure you can think of better ones.

*- We added Test 9-10 following the suggestions. In Test 9, we used smoothed monthly mean meteorological data to drive the model simulations. In Test 10, we eliminated the stochasticity in all model parameters and used only the mean values for simulation.*

Continuation of my previous comment: I would also be interested in seeing the effects of the water retention and photosynthetic water-response parameters separately. Especially since the parameters for the latter may suffer from some measurement artefacts.

R: This is a very appreciated comment. Our future goal is also to make the picture clearer and understanding the factorial effects is a very important aspect. At the moment, our data and techniques are insufficient to separate the different effects. Therefore, model testing based on the parameters quantified by the "mixed" information could be less informative, unless we have had improved measurement data.

In addition, S. fallax and S. magellanicum are largely different in both water retention and photosynthetic response to water stress. Further testing on species either with similar water retention, or with similar photosynthetic response would be more informative to this question.

My new comment: In my mind, a model is the perfect opportunity to pretend that your species are not different in both but just in one or the other aspect and to test the individual effects of these parameters. You would not have real data to validate the result, but that is not the point here. The point is to understand what these parameters do and what would happen with hypothetical species with these parameter combinations. For me, this type of test is what would constitute a `systematic assessment of the processes´ (L189 in your response).

*-Based on the suggestion, we simulated the mean cover of the two moss species by setting photosynthetic water-response parameters to be the same for both species but keep the water retention effects different (Test 7-8)*

My new comment regarding my previous point 7: I did not mean to say that module III is unimportant, but it is not evaluated in this paper (just used to create input data for the presented model modules), so it seems too much (therefore unbalanced) to spend several pages on explaining the details of module III. I would recommend moving this information to the supplement.
*- Moved to the supplements (as a part of Supplement A)*

Additionally: Table 4, the sensitivity analysis: how do these 10% changes relate to the actual uncertainty in the parameter values?

-*We used the 10% changes to test the parameter sensitivity, as we have limited information on the actual values of the parameters regarding to our site. Indeed, a change of 10% may not be comparable to the actual uncertainty of some parameter values (e.g. hydraulic conductivity of peat may vary by several orders, depending on the peat quality). However, the test for parameter sensitivity here aims to reveal the robustness of model at "current" states, rather than to investigate scenarios with different actual variations.*

[revised manuscript text omitted]